# Hepatic conversion of acetyl-CoA to acetate plays crucial roles in energy stress

Jinyang Wang[1†], Yaxin Wen[1†], Wentao Zhao[1†], Yan Zhang[1], Furong Lin[1], Cong Ouyang[1], Huihui Wang[1], Lizheng Yao[1], Huanhuan Ma[1], Yue Zhuo[1], Huiying Huang[1], Xiulin Shi[2], Liubin Feng[3], Donghai Lin[4]*, Bin Jiang[1]*, Qinxi Li[1]*

[1]State Key Laboratory of Cellular Stress Biology, School of Life Sciences, Faculty of Medicine and Life Sciences, Xiamen University, Xiamen, China; [2]Department of Endocrinology and Diabetes, Xiamen Diabetes Institute, Fujian Province Key Laboratory of Translational Research for Diabetes, The First Affiliated Hospital of Xiamen University, Xiamen, China; [3]High-Field NMR Center, College of Chemistry and Chemical Engineering, Xiamen University, Xiamen, China; [4]Department of Chemical Biology, College of Chemistry and Chemical Engineering, Xiamen University, Xiamen, China

*For correspondence:
dhlin@xmu.edu.cn (DL);
jiangbin@xmu.edu.cn (BJ);
liqinxi@xmu.edu.cn (QL)

†These authors contributed equally to this work

Competing interest: The authors declare that no competing interests exist.

**Abstract** Accumulating evidence indicates that acetate is increased under energy stress conditions such as those that occur in diabetes mellitus and prolonged starvation. However, how and where acetate is produced and the nature of its biological significance are largely unknown. We observed overproduction of acetate to concentrations comparable to those of ketone bodies in patients and mice with diabetes or starvation. Mechanistically, ACOT12 and ACOT8 are dramatically upregulated in the liver to convert free fatty acid-derived acetyl-CoA to acetate and CoA. This conversion not only provides a large amount of acetate, which preferentially fuels the brain rather than muscle, but also recycles CoA, which is required for sustained fatty acid oxidation and ketogenesis. We suggest that acetate is an emerging novel 'ketone body' that may be used as a parameter to evaluate the progression of energy stress.

## eLife assessment

This is **important** work that examines hepatic acetate production via ACOT12/18 in starvation and diabetes. The investigators use **solid** loss of function strategies in cells, including mouse primary hepatocytes, and in vivo mouse experiments to show that ACOTs are necessary for normal acetate production in the context of fasting and type 1 diabetes. Given that acetate is commonly thought to primarily represent a fermentation product, this study is of interest as it describes hepatic pathways converting fatty acids to acetate.

## Introduction

Disordered homeostasis of energy metabolism, which is associated with emergency situations such as untreated diabetes mellitus, prolonged starvation, and ischemic heart/brain disease, is a serious threat to human health (*Field et al., 2001*; *Galgani and Ravussin, 2008*; *Martinic and von Herrath, 2008*; *Must et al., 1999*). In response to such disorder, the metabolic patterns of multiple organs have to be remodeled to rescue the imbalance and bring whole organism through the crisis (*Denechaud et al., 2008*; *Frühbeck et al., 2001*; *Goldberg et al., 2018*; *Hirai et al., 2021*; *Meier and Gressner,*

*2004*; *Nishimoto et al., 2016*; *Palikaras et al., 2015*; *Russell and Cook, 1995*). Ketone bodies, namely acetoacetate (AcAc), β-hydroxybutyrate (3-hydroxybutyrate, 3-HB), and acetone, are over-produced from fatty acids in the liver under conditions in which carbohydrate availability is reduced, such as diabetes and starvation. These bodies are released into blood and serve as a vital alternative metabolic fuel for extrahepatic tissues including the brain, skeletal muscle, and heart, where they are converted to acetyl-CoA and oxidized in the tricarboxylic cycle (TCA), providing a large amount of energy (*Cahill, 2006*; *D'Acunzo et al., 2021*; *Dentin et al., 2006*; *Krishnakumar et al., 2008*; *Puchalska and Crawford, 2017*; *Robinson and Williamson, 1980*).

Previous studies have shown that acetate concentrations are significantly increased in diabetes and prolonged starvation (*Akanji et al., 1989*; *Seufert et al., 1984*; *Todesco et al., 1993*), and acetate is considered to be a nutrient that nourishes the organism by undergoing conversion to acetyl-CoA which is further catabolized in the TCA (*Lindsay and Setchell, 1976*; *Liu et al., 2018*; *Schug et al., 2015*; *Schug et al., 2016*). Conversely, acetyl-CoA can also be hydrolyzed to acetate by proteins of the acyl-CoA thioesterase (ACOT) family (*Swarbrick et al., 2014*; *Tillander et al., 2017*). Unfortunately, it is not clear where, under what conditions and how acetate is produced, nor what is its biological significance. Considering that both acetate and ketone bodies are produced from acetyl-CoA and catabolized back to acetyl-CoA, we thoroughly investigated the production and utilization of acetate, also looking at ketone bodies as a comparison. We suggest that acetate is an emerging novel 'ketone body' that plays important roles, similar to those of classic ketone bodies, in energy stress conditions such as diabetes mellitus and prolonged starvation.

Note: our description of acetate as an emerging novel 'ketone body' does not suggest that it as a real ketone in structure, but emphasizes the high similarity of acetate and classic ketone bodies in terms of both the organ in which they are produced (liver), the substrate from which they are produced (fatty acids-derived acetyl-CoA), the roles they play as important sources of fuel and energy for many extrahepatic peripheral organs, their catabolism back to acetyl-CoA and degradation in the TCA cycle, and the physiological conditions of their production (under energy stresses such as prolonged starvation and untreated diabetes mellitus).

## Results

### Acetate is dramatically elevated under energy stress conditions in mammals

To investigate whether acetate is produced in a pattern similar to ketone bodies, we measured serum glucose, 3-beta-hydroxybutyrate (3HB), acetoacetate (AcAc), and other metabolites in 17 diabetes mellitus patients and 8 healthy volunteers as controls (*Figure 1—figure supplement 1A*; *Figure 1—source data 1*). We observed a significant increase of acetate in parallel with the canonical elevation of ketone bodies (3-HB and AcAc) and serum glucose in diabetes mellitus patients as compared with healthy controls (*Figure 1A*). We then detected acetate in mouse models and found that the levels of serum acetate and ketone bodies were dramatically elevated to the same extent in streptozotocin (STZ)-induced type I diabetic C57BL/6 (*Figure 1B*) and BALB/c mice (*Figure 1—figure supplement 1B*) as in type II diabetic db/db mice (*Figure 1—figure supplement 1C*). As expected, starvation also leads to a marked decrease in serum glucose concentration and an increase in serum acetate and ketone body levels in normal C57BL/6 (*Figure 1C*) and BALB/c mice (*Figure 1—figure supplement 2*). These data demonstrate that serum acetate is boosted to the same extent as canonical ketone bodies under energy stresses including those that occur with diabetes mellitus and starvation. For the sake of simplicity, we will refer to this acetate as 'energy stress-induced acetate (ES-acetate)'.

### ES-acetate is derived from free fatty acids in mammalian cells

Next, we asked which nutrients ES-acetate is derived from. In mammals, serum acetate generally has three sources: dietary acetate, as a metabolic product of gut microbiota, and as the intermediate of intracellular biochemical processes (*Schug et al., 2016*). As the mice used in this work were fed an acetate-free diet, we focused on acetate formed by gut microbiota and endogenous biochemical reactions. To determine whether gut microbiota contribute to the production of ES-acetate, mice were pre-treated with antibiotics to eliminate gut microbes (saline as control) as reported previously (*Sivan et al., 2015*). We observed that antibiotic pre-treatment failed to obviously affect the production

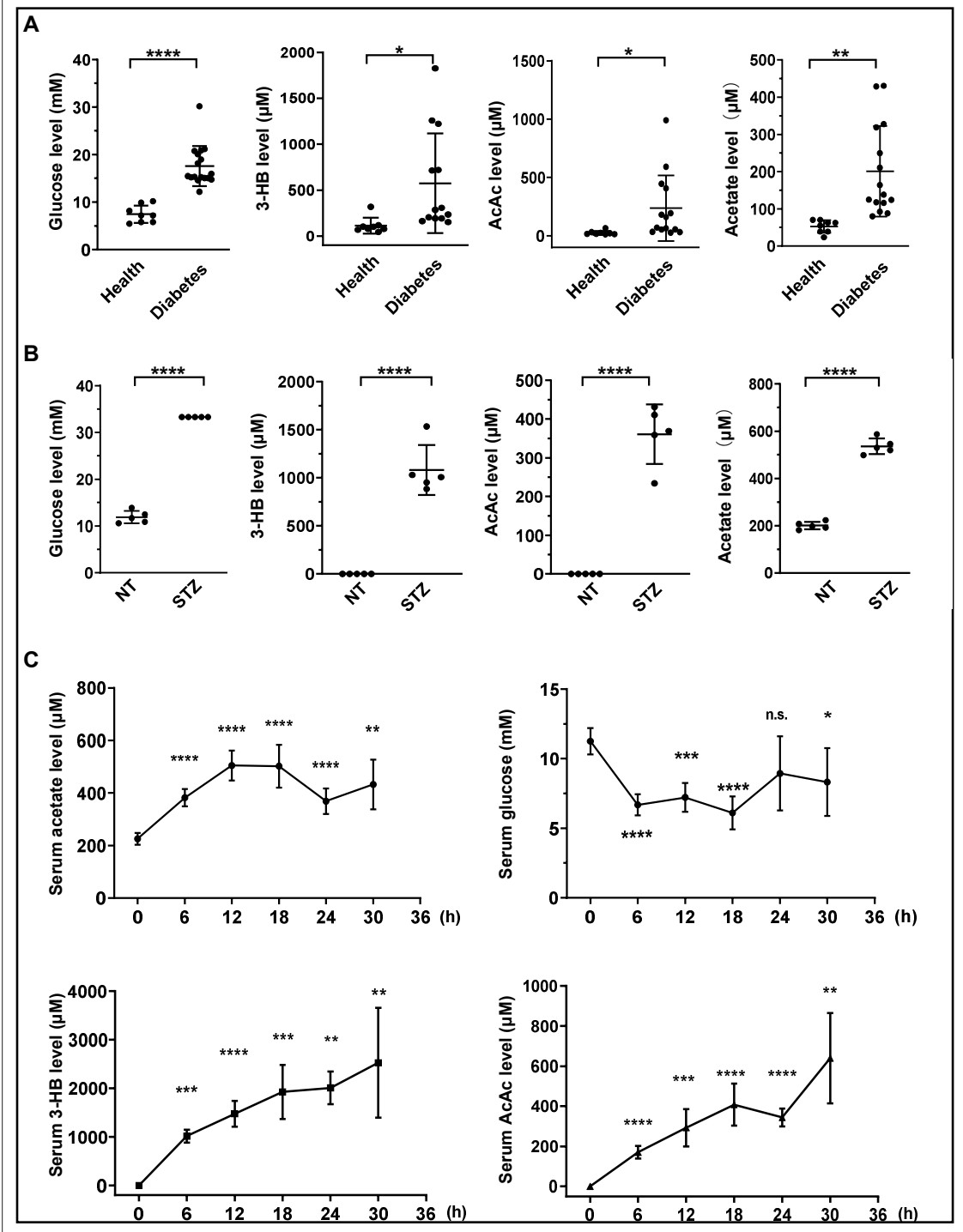

**Figure 1.** Acetate is produced at a levels comparable to those of ketone bodies in energy stress conditions. (**A**) Enrichment of glucose, 3-HB, AcAc, and acetate in clinical serum samples from healthy volunteers and patients with diabetes mellitus (Healthy, *n* = 8; Diabetes, *n* = 17). (**B**) Enrichment of glucose, 3-HB, AcAc, and acetate in the serum of STZ-induced diabetic mice (C57BL/6, *n* = 5). (**C**) The levels of acetate, 3-HB, AcAc, and glucose in the serum of C57BL/6 mice (*n* = 5) starved for the indicated time course. Abbreviations: 3-HB, 3-hydroxybutyrate; AcAc, acetoacetate; NT, untreated control; STZ, streptozotocin. Values are expressed as mean ± standard deviation (SD) and analyzed statistically by two-tailed unpaired Student's *t*-test (*p < 0.05, **p < 0.01, ***p < 0.001, ****p < 0.0001, n.s., no significant difference).

The online version of this article includes the following source data and figure supplement(s) for figure 1:

**Source data 1.** Information describing the patient and healthy volunteers for the clinical data depicted in *Figure 1*.

*Figure 1 continued on next page*

*Figure 1 continued*

**Figure supplement 1.** Increased concentration of acetate in diabetes mellitus.

**Figure supplement 2.** Increased level of acetate in fasting mice.

of acetate that was induced by either starvation (*Figure 2—figure supplement 1A, B*) or diabetes (*Figure 2—figure supplement 1C*), demonstrating that ES-acetate is mainly produced endogenously. Next we used nuclear magnetic resonance (NMR) (*Figure 2—figure supplement 2A, B*) and gas chromatography–mass spectrometry (GC–MS) (*Figure 2—figure supplement 2C, D*) to detect the acetate that was secreted into the culture medium by several cell lines, and found that these cells showed different ability in producing acetate. Consistently, *Liu et al., 2018* reported that acetate is derived from glucose in mammalian cells supplied with abundant nutrients. Indeed, we observed the secretion of different amounts of U-$^{13}$C-acetate after cells were cultured in medium supplemented with U-$^{13}$C-glucose (*Figure 2—figure supplement 3A*). Interestingly, we also observed the production of 36.6%

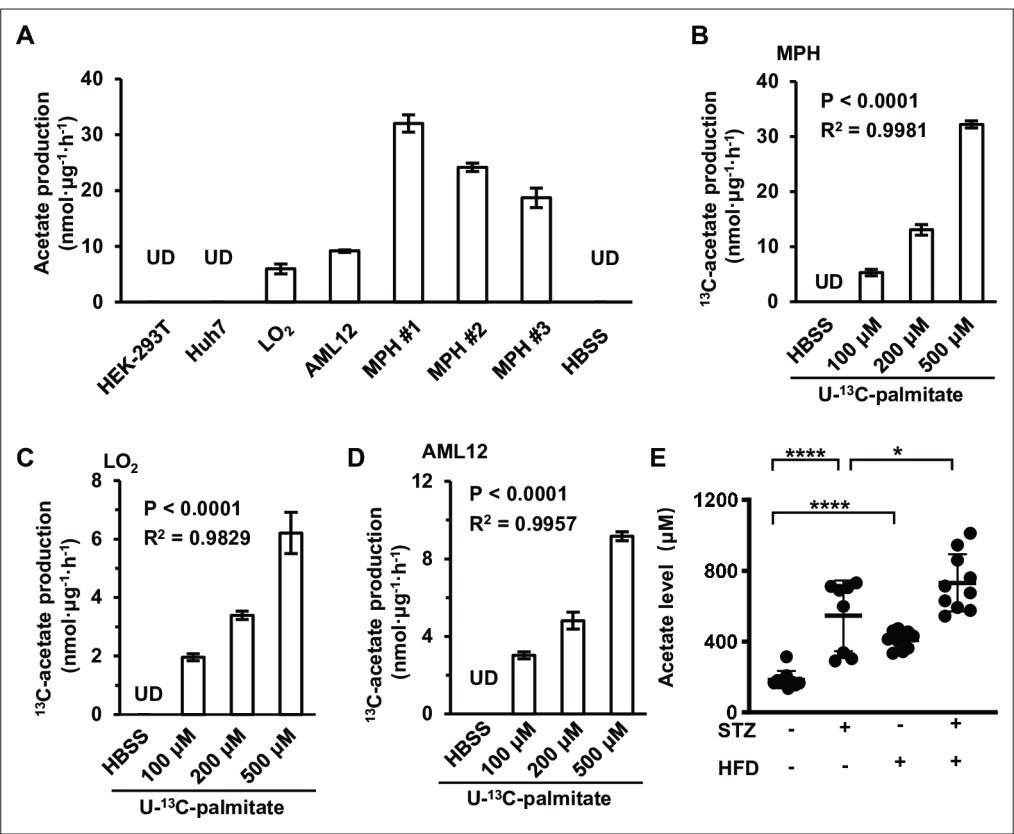

**Figure 2.** Acetate is derived from free fatty acids (FFAs) in mammalian cells. (**A**) The amount of U-$^{13}$C-acetate secreted by the indicated cell lines cultured in U-$^{13}$C-palmitate-containing Hanks' balanced salt solution (HBSS) for 20 hr ($n = 3$). (**B–D**) The amount of U-$^{13}$C-acetate secreted by MPH (**B**), LO$_2$ (**C**), and AML12 (**D**) cells cultured in HBSS supplemented with increasing doses of U-$^{13}$C-palmitate for 20 hr ($n = 3$). (**E**) Enrichment of acetate in the serum of untreated or STZ-induced diabetic C57BL/6 mice ($n = 10$) fed with a high-fat diet (HFD) or a control diet. Abbreviations: MPH, mouse primary hepatocyte; STZ, streptozotocin; UD, undetectable. Values are expressed as mean ± standard deviation (SD) and were analyzed statistically using a two-tailed unpaired Student's *t*-test (**A, E**) or one-way analysis of variance (ANOVA) (**B–D**) (*p < 0.05, ****p < 0.0001, n.s., no significant difference).

The online version of this article includes the following figure supplement(s) for figure 2:

**Figure supplement 1.** Acetate is increased independently of gut microbiota upon energy stress.

**Figure supplement 2.** Acetate is secreted by *in vitro* cultured mammalian cells.

**Figure supplement 3.** Acetate is derived from other nutrients besides glucose.

**Figure supplement 4.** ES-acetate is mainly derived from free fatty acids (FFAs).

of non-U-$^{13}$C-acetate, indicating that this proportion of acetate is derived from nutrients other than glucose (*Figure 2—figure supplement 3B*).

We then examined whether the acetate secreted by cultured cells is derived from amino acids (AAs) and free fatty acids (FFAs) upon starvation. After different cells were cultured in Hanks' balanced salt solutions (HBSS, free of glucose, fatty acids and amino acids) supplemented with FFAs or amino acids for 20 hr, supplementation with FFAs (*Figure 2—figure supplement 4D, E*) rather than amino acids (*Figure 2—figure supplement 4A–C*) significantly increased acetate levels, suggesting a major contribution of FFAs to acetate production. To confirm this observation, a series of widely used cell lines and mouse primary hepatocytes (MPHs) were cultured in HBSS supplemented with U-$^{13}$C-palmitate, before secreted U-$^{13}$C-acetate was measured (*Figure 2A*). These cell lines displayed quite different abilities to convert palmitate to acetate, and were accordingly divided into FFA-derived acetate-producing cells (FDAPCs: LO$_2$, MPH, AML12, etc.) and no-FFA-derived acetate-producing cells (NFDAPCs: HEK-293T, Huh7, etc.). All of the FDAPCs secreted U-$^{13}$C-acetate in a manner that was dependent on the dose of U-$^{13}$C-palmitate supplementation (*Figure 2B–D*). We also observed that high-fat diet induced a significant increase in acetate production in both normal and STZ-induced diabetic mice (*Figure 2E*). Taken together, these findings suggest that acetate can be derived from FFAs in energy stress conditions.

## ACOT12 and ACOT8 are involved in acetate production in mammalian cells

It has been reported that acyl-CoAs with different lengths of carbon chain could be hydrolyzed to FFAs specifically by a corresponding ACOTs family protein (*Tillander et al., 2017*). Acetyl-CoA, the shortest chain of acyl-CoA and the critical product of β-oxidation, is hydrolyzed to acetate by acyl-CoA thioesterase 12 (ACOT12) (*Swarbrick et al., 2014*). We next analyzed the GEO database and found out that the expression of *Acot1/2/8/12* is upregulated significantly alongside the increase of β-oxidation and ketogenesis in mice liver after 24 hr of fasting (*Figure 3A*; *Figure 3—figure supplement 1A*). To determine which ACOT is responsible for ES-acetate production, we overexpressed a series of ACOTs in HEK-293T cells and observed large amount of acetate production when either ACOT8 or ACOT12 was overexpressed (*Figure 3B*), indicating that these two ACOTs are involved in ES-acetate production. Consistently, the protein levels of both Acot12 and Acot8 are upregulated robustly in the livers of either starved mice or STZ-induced type I diabetic mice (*Figure 3—figure supplement 1B, C*). Furthermore, when ACOT12 and ACOT8 were separately overexpressed in NFDAPCs HEK-293T and Huh7, FFA-derived acetate was significantly increased (*Figure 3—figure supplement 1D, E*). Similarly, overexpression of wildtype ACOT12 and ACOT8, rather than their enzyme activity-dead mutants, in HEK-293T (*Figure 3C*) and Huh7 (*Figure 3D*) cells drastically increased the production of U-$^{13}$C-acetate derived from U-$^{13}$C-palmitate (*Ishizuka et al., 2004*; *Swarbrick et al., 2014*). By contrast, knockdown (KD) of ACOT12 or ACOT8 in FDAPCs MPH (*Figure 3E, F*) and LO$_2$ (*Figure 3—figure supplement 1F, G*) diminished U-$^{13}$C-acetate production. These data reveal that ACOT12 and ACOT8 are responsible for ES-acetate production.

## Hepatic ACOT12 and ACOT8 are responsible for ES-acetate production in energy stress conditions

Next we were prompted to determine which organ and subcellular structures are mainly involved in the generation of ES-acetate. First, we analyzed the expression of ACOTs individually at mRNA level in various tissues of human and mice by employing the GTEx and GEO databases. *ACOT12* is mainly expressed in human liver together with genes encoding ketogenic enzymes (*HMGCS2*, *HMGCSL*, *ACAT1*, and *BDH1*), whereas *ACOT8* is expressed ubiquitously at a relative high level in most tissues (*Figure 4—figure supplement 1*). *Acot12* is also expressed mainly in mouse liver and kidney, whereas *Acot8* seems to be expressed at a much lower level in nearly all of the mouse tissues examined (*Figure 4—figure supplement 2A*). Differing from their mRNA expression patterns in the GEO database, we observed high levels of both Acot12 and Acot8 proteins in mouse liver and kidney (*Figure 4—figure supplement 2B*). Consistently, adenovirus-mediated liver-targeted knockdown of either Acot12 or Acot8 dramatically abolished acetate production by starved or diabetic C57BL/6 mice (*Figure 4A–F*), and conditional deletion of Acot12 or Acot8 in liver dramatically decreased acetate production in starved mice (*Figure 4G–J*), demonstrating that the liver is the main organ responsible for ES-acetate production. Moreover, U-$^{13}$C-acetate derived from U-$^{13}$C-palmitate in glucose-free

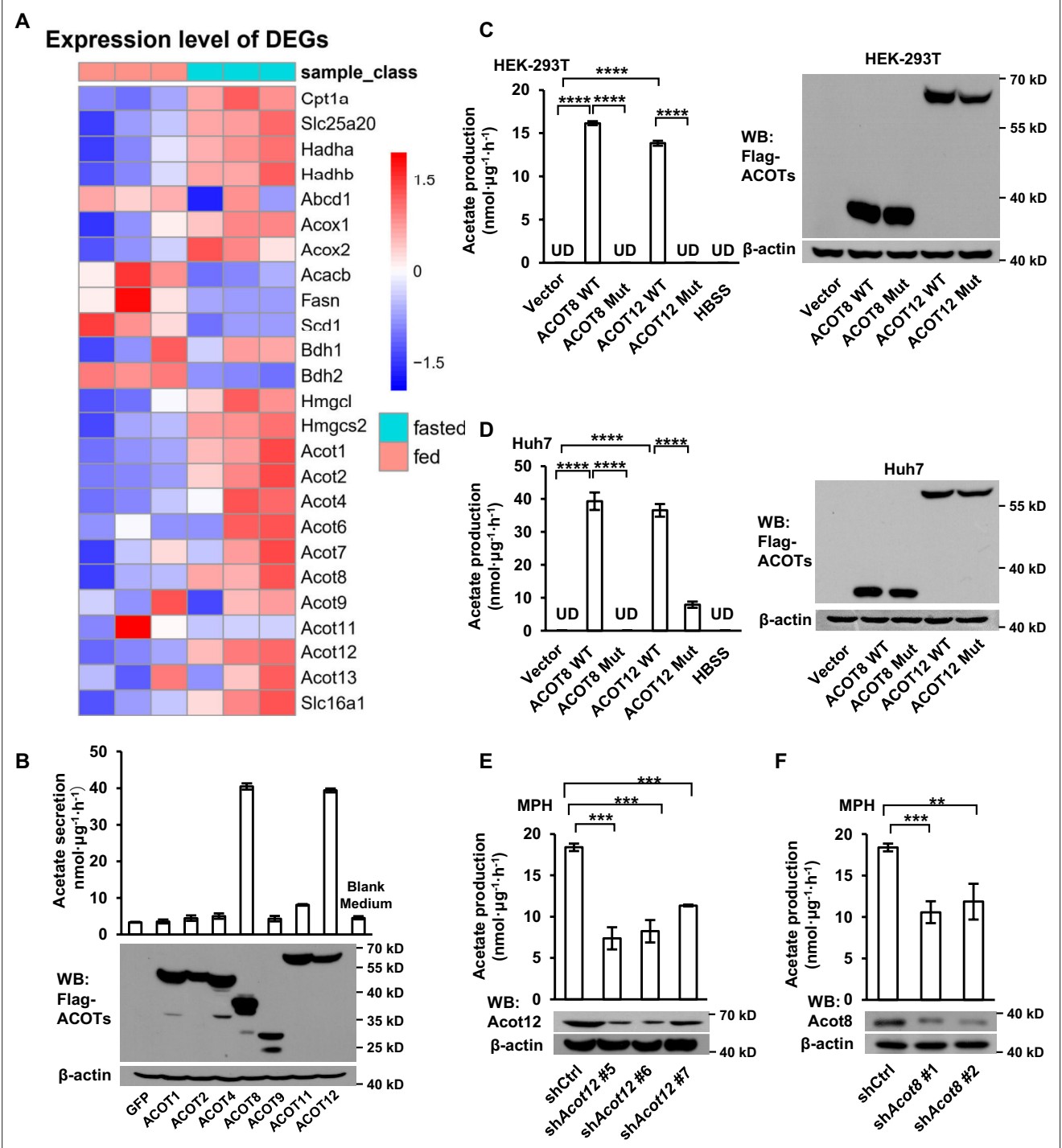

**Figure 3.** ACOT12 and ACOT8 are involved in acetate production in mammalian cells. (**A**) Heatmap showing the differential hepatic expression of genes in the fed and fasted groups. RNAseq analysis data from *Goldstein et al., 2017*. (**B**) The secretion of acetate (upper panel) by HEK-239T cell lines overexpressing various ACOTs, and the protein levels of expressed ACOTs (lower panel). (**C,D**) HEK-293T (**C**) and Huh7 (**D**) cell lines overexpressing control vector, wildtype (WT) ACOT12 and ACOT8 or enzyme activity-dead mutants (Mut) of these two enzymes were cultured in Hanks' balanced salt solution (HBSS) containing U-[13]C-palmitate for 20 hr, before U-[13]C-acetate was detected. (**E, F**) U-[13]C-acetate secreted by Acot12- or Acot8-knockdown mouse primary hepatocytes (MPH) after incubation in U-[13]C-palmitate-containing HBSS for 20 hr. Abbreviations: ACOT8 Mut, ACOT8 H78A mutant; ACOT12 Mut, ACOT12 R312E mutant; sh*Acot8*, short hairpin RNA targeting mouse *Acot8* gene; sh*Acot12*, short hairpin RNA targeting mouse *Acot12* gene; UD, undetectable. Values are expressed as mean ± standard deviation (SD) (*n* = 3) of three independent experiments and analyzed using unpaired Student's *t*-tests (**p < 0.01, ***p < 0.001, ****p < 0.0001, n.s., no significant difference).

*Figure 3 continued on next page*

*Figure 3 continued*

The online version of this article includes the following source data and figure supplement(s) for figure 3:

**Source data 1.** Complete, unedited immunoblots, as well as immunoblots including sample and band identification, are provided for the immunoblots presented in *Figure 3*.

**Figure supplement 1.** ACOT12 and ACOT8 are responsible for acetate production during energy stress.

**Figure supplement 1—source data 1.** Complete, unedited immunoblots, as well as immunoblots including sample and band identification, are provided for the immunoblots presented in *Figure 3—figure supplement 1*.

HBSS was diminished by replenishment of glucose (*Figure 4—figure supplement 3*), in accordance with the concept that glucose is preferable to fatty acids as an energy source. These observations demonstrate that hepatic ACOT12 and ACOT8 are induced and responsible for ES-acetate production in diabetes mellitus and during starvation.

## ACOT12- and ACOT8-catalyzed acetate production is dependent on the oxidation of FFA in both mitochondria and peroxisomes

When next made efforts to identify the subcellular domains in which acetate is produced. Immuno-fluorescence staining and cell fractionation showed that ACOT12 was largely localized in cytosol and ACOT8 mainly in peroxisome (*Figure 5A, B*). It is well known that fatty acids of different chain lengths can be oxidized to yield acetyl-CoA in either the mitochondria or peroxisomes of hepatocytes, and that mitochondrial acetyl-CoA produced in fatty acid oxidation (FAO) is often exported to the cytosol in the form of citrate, which is further cleaved back to acetyl-CoA by ATP citrate lyase (ACLY) (*Figure 5H*; *Lazarow, 1978*; *Leighton et al., 1989*; *Lodhi and Semenkovich, 2014*). Thus, we examined acetate production after mitochondria- or peroxisome-yielded acetyl-CoA had been blocked. Knockdown or etomoxir inhibition of carnitine palmitoyltransferase 1 (CPT1), the main mitochondrial fatty acids transporter, decreased more than one-half of U-$^{13}$C-palmitate-derived U-$^{13}$C-acetate production in LO$_2$ cell lines, despite mitochondrial β-oxidation being almost completely abolished (*Figure 5C–E*). Similarly, knockdown of ACLY diminished palmitate-derived acetate production to the same extent as CPT1 KD (*Figure 5F*). We then knocked down ATP-binding cassette subfamily D member 1 (ABCD1), a peroxisome fatty acids transporter, and observed a less than one-half decline in the production of $^{13}$C-palmitate-derived U-$^{13}$C-acetate (*Figure 5G*). These results, together with the localization of ACOT12 and ACOT8, suggest that acetyl-CoA produced in FAO in the mitochondria and peroxisome is converted to acetate in the cytosol by ACOT12 and in peroxisomes by ACOT8 (*Figure 5H*).

## ACOT12- and ACOT8-catalyzed recycling of CoA from acetyl-CoA is crucial for sustainable FAO

Subsequently, we tried to explore the biological significance of ES-acetate production in response to energy stress by detecting a series of serum metabolic parameters. Knockdown of Acot12 or Acot8 failed to alter the levels of blood glucose as well as insulin in fasted and non-fasted mice, implying that these two molecules may not be involved in glucose metabolism (*Figure 6—figure supplement 1A–C*). However, knockdown of these enzymes caused significant accumulation of total FFAs and various saturated or unsaturated fatty acids, whereas triacylglycerol (TG) levels were not altered (*Figure 6—figure supplement 1D–J*). Considering that in diabetes and after prolonged starvation mobilized lipid is mainly transported in the form of plasma albumin-bound fatty acids, rather than TG, these results suggest that ACOT12 and ACOT8 might be required for rapid degradation of fatty acids in these cases. Indeed, we detected attenuated FAO in ACOT12 and ACOT8 knockdown MPH and LO$_2$ cells (*Figure 6A, B*; *Figure 6—figure supplement 1K, L*). We were then prompted to identify the mechanism underlying such attenuation of FAO. A clue is the knowledge that reduced free Coenzyme A (CoA) is a crucial coenzyme for many metabolic reactions, including those involved in oxidative degradation of fatty acid. Maintenance of the balance between the reduced CoA pool and the oxidized CoA pool is definitely important for sustaining those reactions (*Sivanand et al., 2018*). We wanted to know whether ACOT12- and ACOT8-catalyzed conversion of acetyl-CoA to free CoA plays a key role in maintaining both free CoA level and the balance between reduced CoA and oxidized CoA. To our surprise, in Acot12 or Acot8 KD MPHs, the level of reduced CoA was decreased by 75.2% and 68.3%, respectively; acetyl-CoA increased by 3.49- and 1.71-fold, respectively; and the ratios of

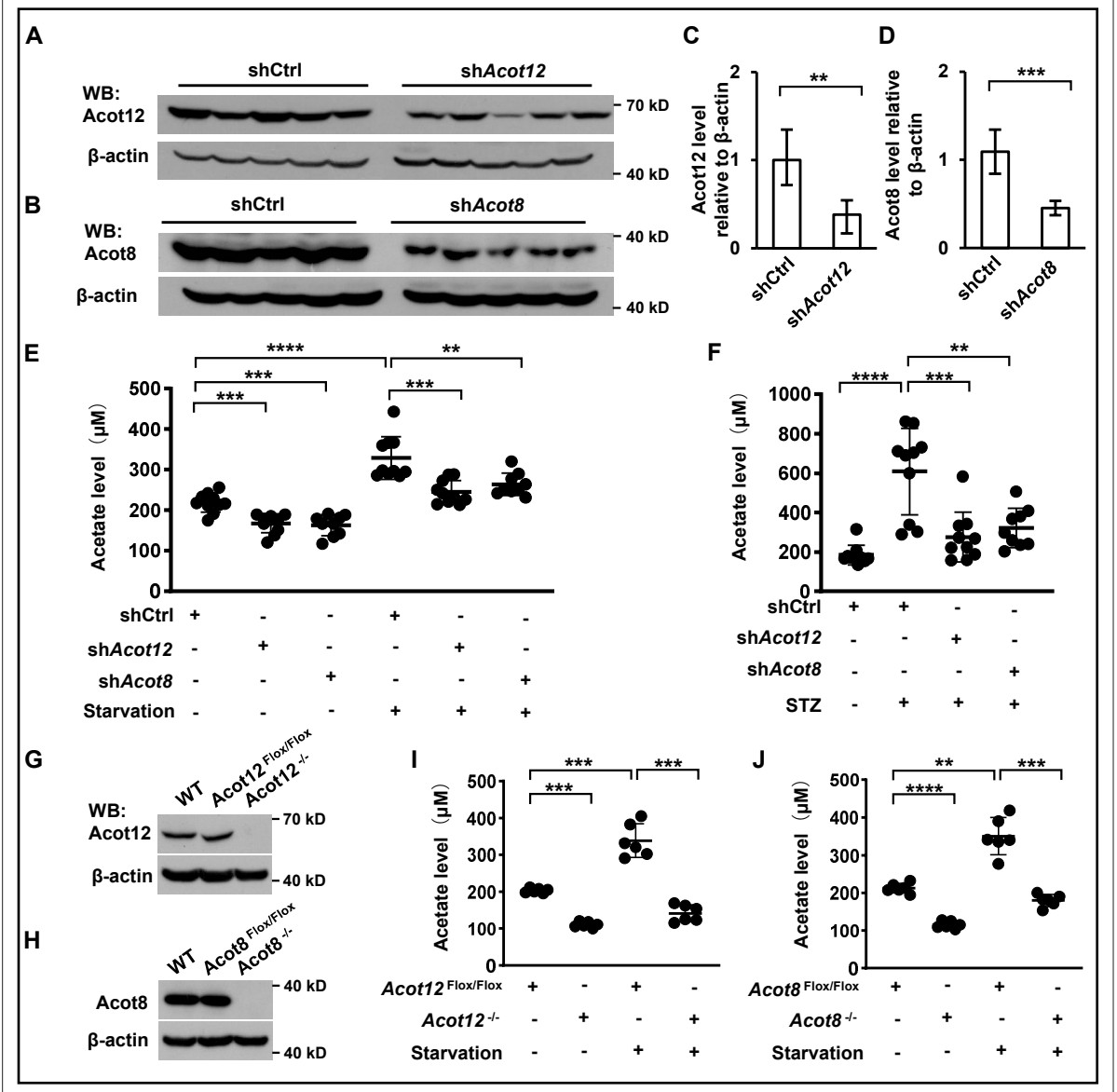

**Figure 4.** ACOT12 and ACOT8 are responsible for acetate production in energy stress conditions. (**A, C**) Acot12 in mice (C57BL/6) liver was knocked down by adenovirus-based shRNA, followed by detection of Acot12 protein with Western Blot (**A**) and evaluation of knockdown efficiency by calculating Acot12 level relative to β-actin (**C**). (**B, D**) The knockdown efficiency of Acot8 was determined in the same way as that of Acot12. (**E**) Enrichment of serum acetate in normal diet and 16 hr fasted mice (C57BL/6) with adenovirus-mediated knockdown of Acot12 or Acot8 in the liver. (**F**) Enrichment of serum acetate in streptozotocin (STZ)-induced diabetic mice (C57BL/6) with adenovirus-mediated knockdown of Acot12 or Acot8 in the liver. (**G, H**) Acot12 (**G**) or Acot8 (**H**) was conditionally deleted in the liver of mice (C57BL/6) by Cre-Loxp in liver, followed by detection of Acot12 and Acot8 protein with Western Blot. (**I, J**) Enrichment of serum acetate in normal diet and 16 hr fasted mice (C57BL/6) with Cre-Loxp-mediated conditional deletion of Acot12 (**I**) or Acot8 (**J**) in liver. Results are expressed as mean ± standard deviation (SD) of three independent experiments in (**C, D**), $n = 10$ mice per group in (**E, F**) and $n = 6$ mice per group in (**I, J**). Results were analyzed by unpaired Student's $t$-tests (**p < 0.01, ***p < 0.001, ****p < 0.0001, n.s., no significant difference).

The online version of this article includes the following source data and figure supplement(s) for figure 4:

**Source data 1.** Complete, unedited immunoblots, as well as immunoblots including sample and band identification, are provided for the immunoblots presented in *Figure 4*.

**Figure supplement 1.** The expression profile of ACOTs and ketogenetic enzymes in human liver.

**Figure supplement 2.** The expression profile of ACOTs and ketogenetic enzymes in mouse liver.

**Figure supplement 2—source data 1.** Complete, unedited immunoblots, as well as immunoblots including sample and band identification, are provided for the immunoblots presented in *Figure 4—figure supplement 2*.

**Figure supplement 3.** Free fatty acid (FFA)-derived acetate is diminished by supplementation of glucose.

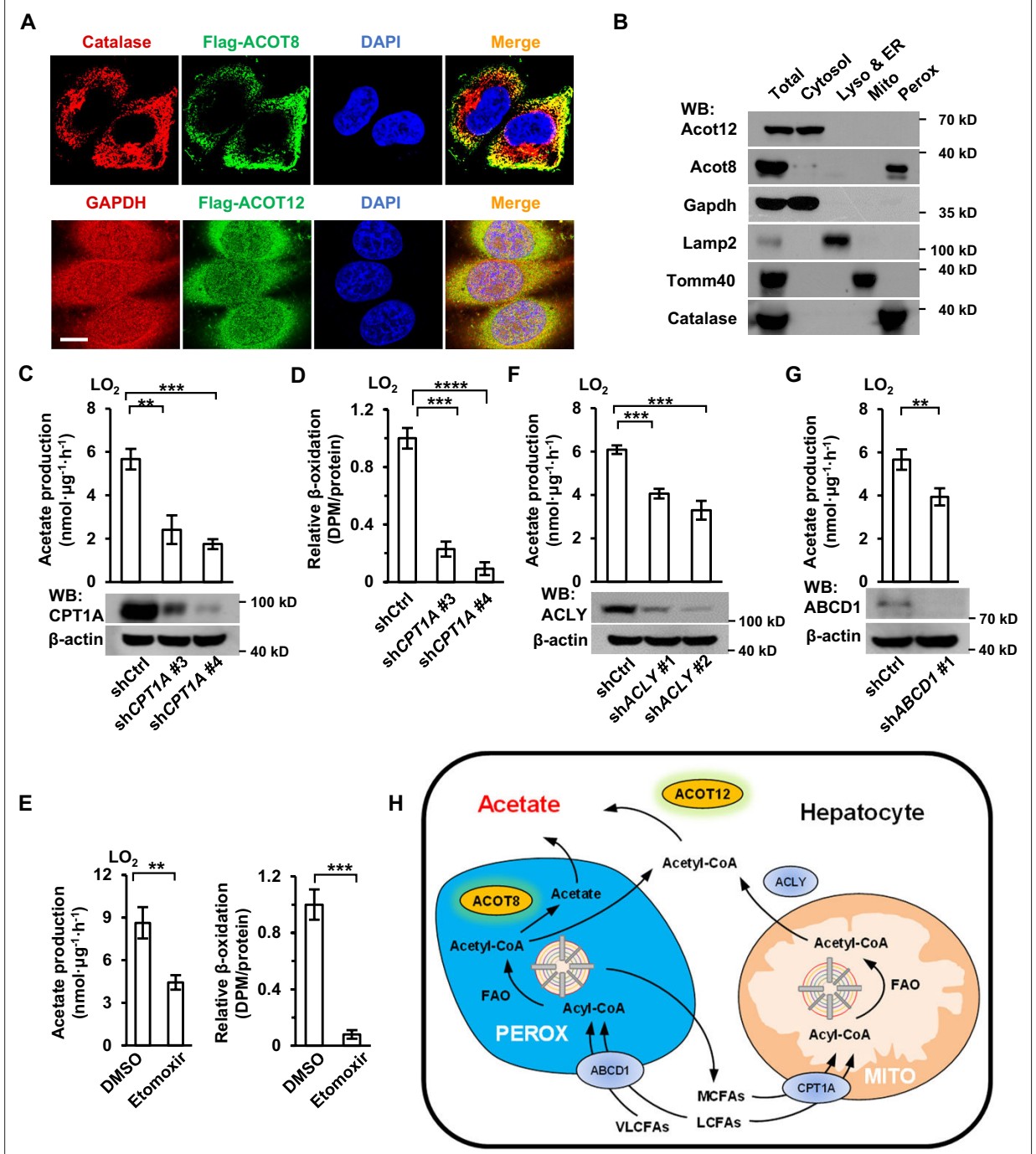

**Figure 5.** Acetate production is dependent on the oxidation of free fatty acids (FFAs) in both mitochondria and peroxisomes. (**A**) Co-immunostaining of Flag-ACOT8 with the peroxisome marker catalase and of Flag-ACOT12 with the cytosol marker GAPDH in LO$_2$ cells. Nuclei were stained with DAPI. Scale bars represent 10 μm. (**B**) The protein levels of Acot12 and Acot8 in the subcellular fractions of mouse primary hepatocyte (MPH) cells. Abbreviations: ER, endoplasmic reticulum; Lyso, lysosome; Mito, mitochondria; Perox, peroxisome. (**C, D**) U-$^{13}$C-acetate production (**C**) and the relative β-oxidation rate (**D**) in carnitine palmitoyltransferase 1A (CPT1A)-knockdown LO$_2$ cells cultured in Hanks' balanced salt solution (HBSS) containing U-$^{13}$C-palmitate for 20 hr. (**E**) U-$^{13}$C-acetate production (left) and the relative β-oxidation rate (right) of LO$_2$ cells cultured in U-$^{13}$C-palmitate-containing HBSS with or without the CPT1 inhibitor etomoxir (20 μM) for 20 hr. (**F**) U-$^{13}$C-acetate production in ATP citrate lyase (ACLY)-knockdown LO$_2$ cells cultured in HBSS supplemented with U-$^{13}$C-palmitate for 20 hr. (**G**) U-$^{13}$C-acetate production in ATP-binding cassette subfamily D member 1 (ABCD1)-knockdown LO$_2$ cells cultured in HBSS containing U-$^{13}$C-palmitate for 20 hr. (**H**) A schematic diagram depicting the mitochondrial and peroxisome pathways of acetate production via the oxidation of FFAs in hepatocytes. Very long- and long-chain fatty acids (VL/LCFAs) are transported through ABCD1 into a peroxisome, where they are further degraded into medium-chain fatty acids (MCFAs) via the fatty acid oxidation (FAO) process. This process involves

*Figure 5 continued on next page*

*Figure 5 continued*

the production of acetyl-CoA, which is further converted to acetate by peroxisome-localized ACOT8. MCFAs generated in peroxisomes are exported into the cytosol and absorbed directly by mitochondria. Cytosolic acyl-CoA derived from medium- and long-chain fatty acids (M/LCFAs) is transferred into mitochondria through CPT1A. All of the fatty acids and acyl-CoA in mitochondria undergo FAO to be degraded to acetyl-CoA. Then, acetyl-CoA together with oxaloacetate is synthesized to form citrate in the tricarboxylic cycle (TCA). Citrate is subsequently exported into the cytosol, where it is lysed to acetyl-CoA by ACLY. Acetyl-CoA is finally converted to acetate by cytosol-localized ACOT12. Values in (**C–G**) are expressed as the mean ± standard deviation (SD) (*n* = 3) of three independent measurements. **p < 0.01, ***p < 0.001, ****p < 0.0001 by two-tailed unpaired Student's *t*-tests.

The online version of this article includes the following source data for figure 5:

**Source data 1.** Complete, unprocessed immunoblots displaying sample and band identification are presented in **Figure 5**, along with the corresponding raw data for immunostaining.

reduced CoA to acetyl-CoA declined from 7.62 to 0.41 and 0.89, respectively (**Figure 6C–E**). In accordance with such alterations, concentrations of other oxidized CoAs (octanoyl-CoA, caproyl-CoA, and succinyl-CoA), whose generation requires adequate levels of reduced CoA as coenzyme, were diminished (**Figure 6—figure supplement 2A–C**). By contrast, metabolites (acetoacetyl-CoA, cholesterol [CHOL], high-density lipoprotein cholesterol [HDL-C], low-density lipoprotein cholesterol [LDL-C]) that have acetyl-CoA as the direct substrate for their synthesis were markedly increased (**Figure 6—figure supplement 2D–G**). It is important to point out that among all of the oxidized CoA compounds examined, the level of acetyl-CoA is far higher than than that of the other molecules, and the switch between acetyl-CoA and reduced CoA plays a significant role in the regulation of CoA pool balance (**Figure 6F**; **Figure 6—figure supplement 2H**). This observation explains why ACOT12- and ACOT8-catalyzed hydrolysis of acetyl-CoA to free CoA and acetate is the crucial step in the maintenance of reduced CoA level and sustained FAO.

## Hydrolysis of acetyl-CoA by ACOT12 and ACOT8 is beneficial to ketogenesis

Distinct from other oxidized CoA compounds, HMG-CoA, a key intermediate for the synthesis of ketone bodies using acetyl-CoA as a substrate, was dramatically reduced in Acot12 and Acot8 KD MPHs, demonstrating that Acot12 and Acot8 may be positive regulators of HMG-CoA level (**Figure 7A**). Accordingly, the main ketone bodies AcAc and 3-HB were decreased significantly in STZ-induced diabetic mice with knockdown of Acot12 or Acot8 (**Figure 7B, C**). To clarify the mechanism that underlies ACOT12/8 regulation of HMG-CoA, we detected the protein level of 3-hydroxy-3-methylglutaryl-CoA synthase 2 (HMGCS2), the key enzyme for HMG-CoA synthesis. Interestingly, Hmgcs2 was remarkably downregulated in Acot12/8 KD MPHs (**Figure 7D–G**), indicating that Acot12 and Acot8 are positive regulators of Hmgcs2 protein level. A previous study showed that HMGCS2 activity is suppressed by acetylation (**Wang et al., 2019**). Thus, we examined the acetylation of Hmgcs2 and observed a clear increase in Hmgcs2 acetylation in Acot12/8 KD MPHs (**Figure 7H**), corresponding to the increase in acetyl-CoA level (**Figure 6D**), the direct substrate of acetylation. This observation demonstrates that ACOT12 and ACOT8 are also positive regulators of HMGCS2 activity by hydrolyzing acetyl-CoA to avoid accumulation of acetyl-CoA and over-acetylation of HMGCS2. Taken together, these results suggest that ACOT12 and ACOT8 are upregulated upon energy stress, and in turn enhance the function of HMGCS2 by increasing not only its concentration but also its activity, facilitating the production of ketone bodies to fuel the extrahepatic tissues.

## Acetate is beneficial to extrahepatic tissues during energy stress

Many studies have shown that the ketone bodies are produced mainly in the liver in diabetes mellitus and prolonged starvation, providing fuel for crucial extrahepatic organs such as the brain (**Puchalska and Crawford, 2017**; **Robinson and Williamson, 1980**). Given that acetate has also been reported to serve as an energy substrate for cells (**Comerford et al., 2014**; **Mashimo et al., 2014**; **Schug et al., 2016**), we wanted to know whether ES-acetate plays the same role as ketone bodies in the same emergency status. 2-$^{13}$C-acetate was injected into starved or STZ-induced diabetic mice intraperitoneally, before $^{13}$C-labeled metabolic intermediates were analyzed by liquid chromatography–mass spectrometry (LC–MS). $^{13}$C-acetyl-CoA and $^{13}$C-incorporated TCA cycle metabolites such as citrate, aconitate, isocitrate, succinate, fumarate, and malate were dramatically increased in the brain (**Figure 8A–G**), but decreased in muscle (**Figure 8—figure supplement 1**), of starved or diabetic mice as compared with

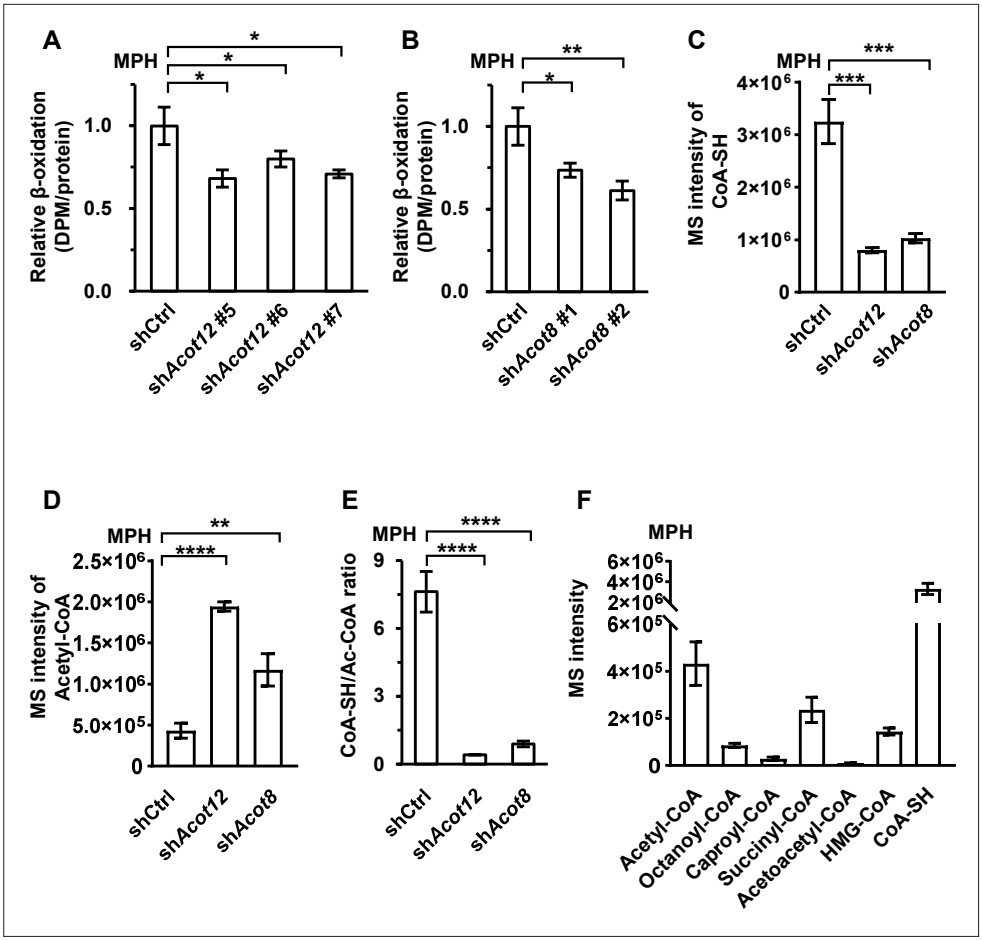

**Figure 6.** ACOT12 and ACOT8 serve to maintain the CoA pool for sustained fatty acid oxidation (FAO). (**A, B**) Mouse primary hepatocytes (MPHs) knocked down for Acot12 (**A**) or Acot8 (**B**) were cultured in glucose-free reaction buffer containing 0.8 μCi/ml [9,10-³H(N)]-oleic acid for 20 hr, before the relative β-oxidation rate was determined. (**C**) Relative abundance ($M + 0$) of reduced CoA in MPHs knocked down for Acot12 or Acot8. (**D**) Relative abundance ($M + 0$) of acetyl-CoA in MPHs knocked down for Acot12 or Acot8. (**E**) The ratio of reduced CoA to acetyl-CoA in MPHs knocked down for Acot12 or Acot8. (**F**) Relative abundance ($M + 0$) of reduced CoA and various oxidized CoA compounds in MPHs. Abbreviations: Ac-CoA, acetyl-CoA; HMG-CoA, 3-hydroxy-3-methylglutaryl-CoA. Values are expressed as mean ± standard deviation (SD) ($n = 3$) of three independent experiments and were analyzed using unpaired Student's $t$-tests (*$p < 0.05$, **$p < 0.01$, ***$p < 0.001$, ****$p < 0.0001$, n.s., no significant difference).

The online version of this article includes the following figure supplement(s) for figure 6:

**Figure supplement 1.** ACOT12 and ACOT8 are involved in the catabolism of fatty acids.

**Figure supplement 2.** ACOT12 and ACOT8 serve to maintain the CoA pool for sustained fatty acid oxidation (FAO).

untreated control mice. This is in line with the notion that the brain has priority in energy expenditure in animals experiencing energy stresses. Moreover, we performed intraperitoneal injection of both acetate and 3-HB simultaneously in fasting mice and compared the serum concentration curves for these compounds. Interestingly, acetate not only took less time to reach peak plasma level than 3-HB (5 vs 12 min), but also much less time to be eliminated (20 vs 120 min), implying that, as previously reported (**Sakakibara et al., 2009**), acetate may be more rapidly absorbed and consumed than 3-HB by the extrahepatic organs of mice (**Figure 8H**). These results suggest that acetate is an emerging novel 'ketone body' produced in liver from FFA and functioning to fuel extrahepatic organs, in particular brain, in the emergency status such as energy stresses. the brain, in the emergency status such as energy stresses. when in an emergency status such as energy stress.

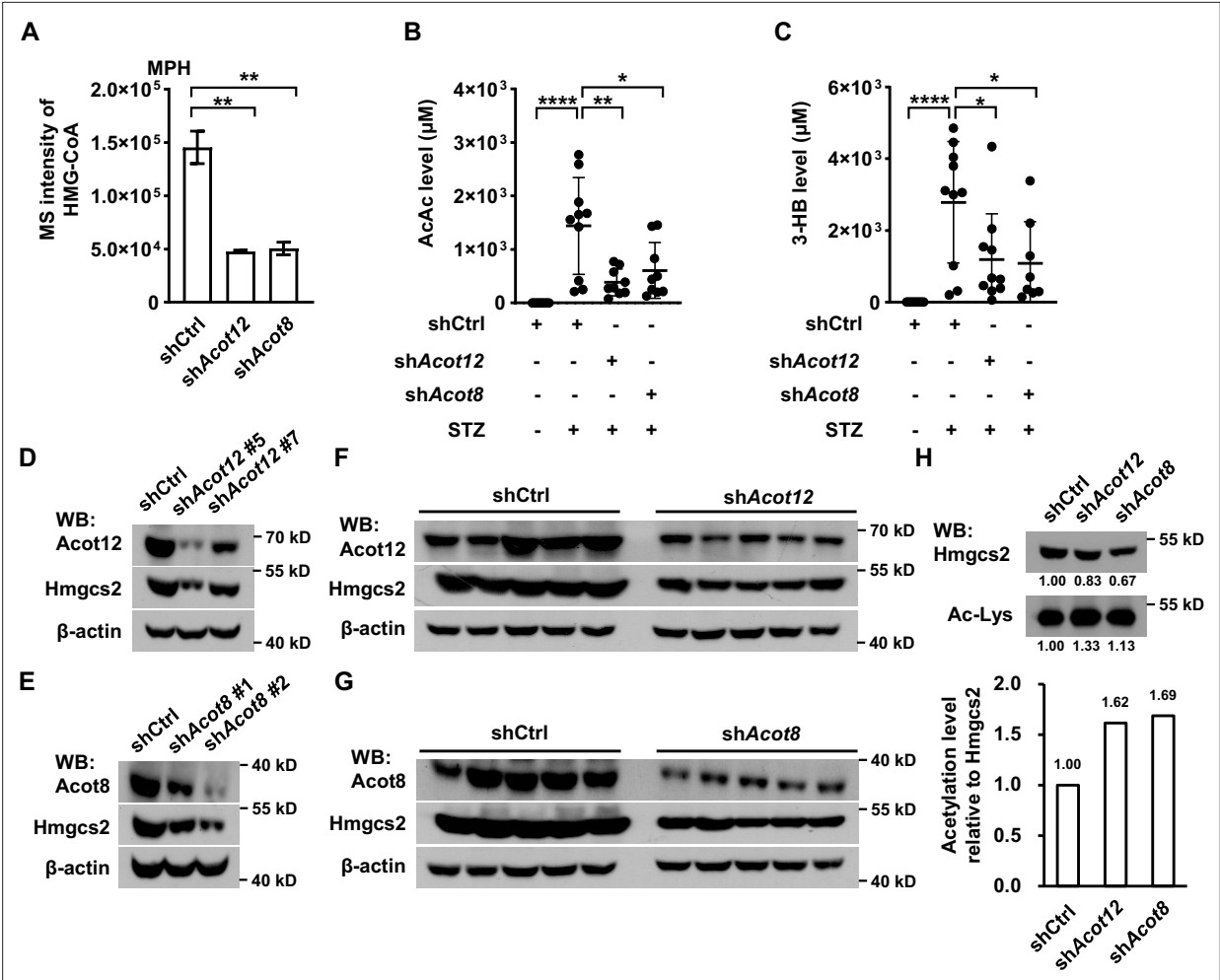

**Figure 7.** ACOT12 and ACOT8 are required for the production of ketone bodies in streptozotocin (STZ)-induced diabetes. (**A**) Relative abundance ($M$ + 0) of HMG-CoA in mouse primary hepatocytes (MPHs) knocked down for Acot12 or Acot8 ($n$ = 3). (**B, C**) Serum levels of acetoacetate (AcAc) (**B**) and 3-hydroxybutyrate (3-HB) (**C**) in STZ-induced diabetic C57BL/6 mice with adenovirus-mediated knockdown of Acot12 or Acot8 in the liver. (**D, E**) The protein levels of Hmgcs2 in MPHs knocked down for Acot12 (**D**) and Acot8 (**E**). (**F, G**) Acot12 (**F**) and Acot8 (**G**) in mice (C57BL/6) liver were knocked down by adenovirus-based shRNA, before Hmgcs2 protein was detected by Western Blot. (**H**) Western Blot (upper panel) and evaluation of the relative acetylation (Ac-Lys) level by calculating the ratio of acetylated Hmgcs2 relative to Hmgcs2 (lower panel). Abbreviations: HMG-CoA, 3-hydroxy-3-methylglutaryl-CoA; Hmgcs2, 3-hydroxy-3-methylglutaryl-CoA synthase 2. Results are expressed as the mean ± standard deviation (SD) of three independent experiments in (**A**) and $n$ = 10 mice per group in (**B, C**), and were analyzed by using unpaired Student's $t$-tests (*$p < 0.05$, **$p < 0.01$, ****$p < 0.0001$, n.s., no significant difference).

The online version of this article includes the following source data for figure 7:

**Source data 1.** Complete, unedited immunoblots, as well as immunoblots including sample and band identification, are provided for the immunoblots presented in *Figure 7*.

Next, we examined the physiological significance of ES-acetate to the behaviors of animals under energy stress. We performed a list of behavioral tests: forelimb grip force test to assess forelimb muscle strength, a rotarod test to examine neuromuscular coordination, an elevated plus maze test (EPMT) to assess anxiety-related behavior, a Y-maze test (YMZT), and a novel object recognition (NOR) test to evaluate working memory and cognitive functions. It is clear that the forelimb strength and running time in the rotarod test were dramatically reduced in diabetic mice, further reduced by knockdown of Acot12 or Acot8, and rescued by administration of exogenous acetate (*Figure 8—figure supplement 2A, B*). Interestingly, the parameters related to muscle force and movement ability in other tests—including total distance in YMZT, total entries in YMZT and total distance in the NOR test—were also decreased in diabetic mice and further worsened by knockdown of Acot12 or Acot8 (*Figure 8—figure supplement 2E, F, H*). These observations demonstrate that ES-acetate is important

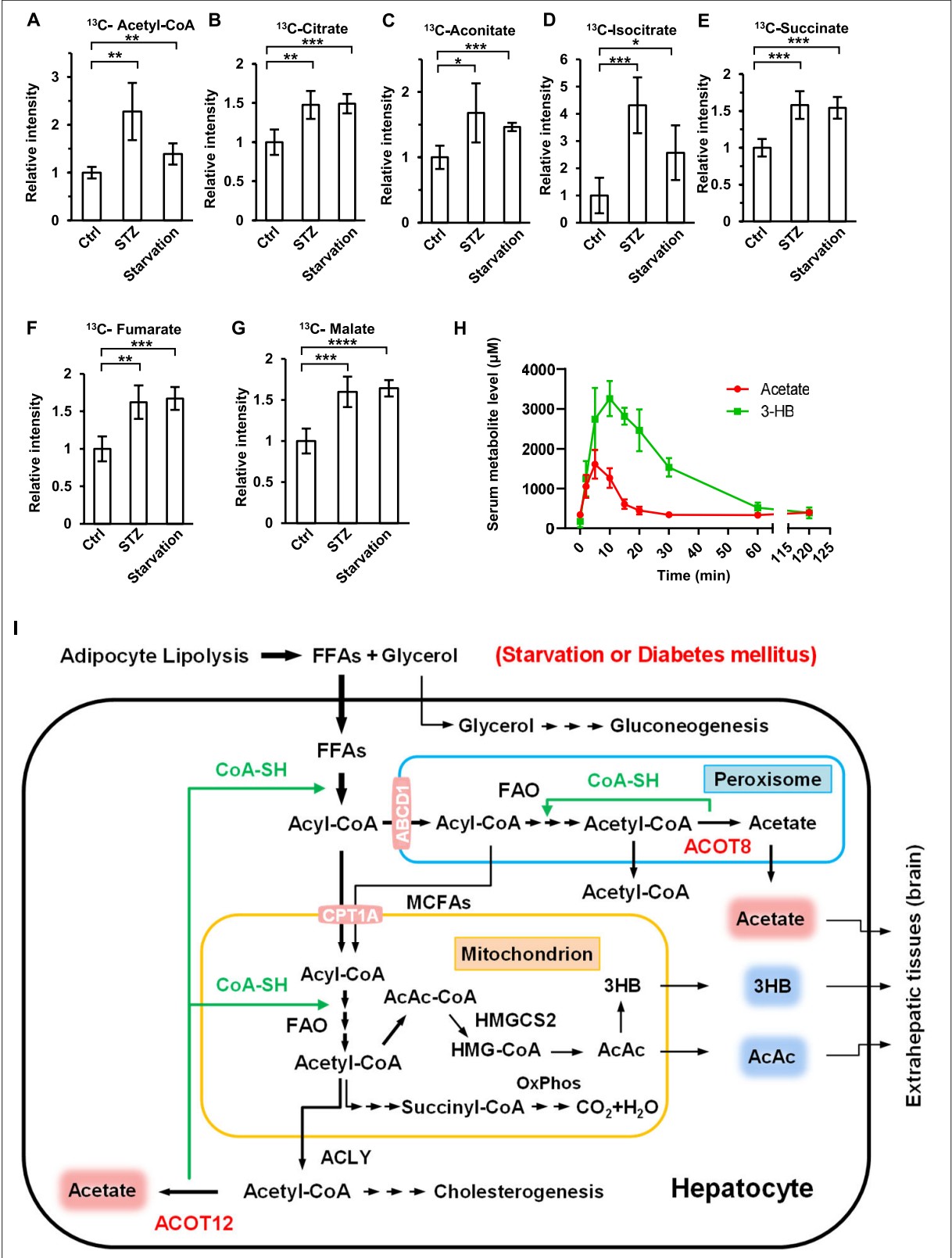

**Figure 8.** Brain exhibits increased acetate consumption during energy stress. (**A–G**) Relative abundances ($M + 1$) of $^{13}$C-acetyl-CoA (**A**), $^{13}$C-citrate (**B**), $^{13}$C-aconitate (**C**), $^{13}$C-isocitrate (**D**), $^{13}$C-succinate (**E**), $^{13}$C-fumarate (**F**), and $^{13}$C-malate (**G**) in the brain of starved or diabetic mice (C57BL/6) were determined 1 hr after intraperitoneal injection of 2-$^{13}$C-acetate (310 mg/kg). (**H**) The abundance of acetate and 3-hydroxybutyrate (3-HB) in the serum of fasting mice (C57BL/6) after intraperitoneal injection (acetate 300 mg/kg, 3-HB 520 mg/kg). (**I**) A working model describing the biological significance

*Figure 8 continued on next page*

*Figure 8 continued*

of ACOT12- and ACOT8-catalized conversion of acetyl-CoA to acetate and CoA. Under energy stress conditions, such as in diabetes mellitus and after prolonged starvation, at least two advantages are obtained by converting acetyl-CoA to acetate and CoA: (1) CoA is required for sustained fatty acid oxidation (FAO) and for the production of ketone bodies in the liver; (2) acetate serves as a novel ketone body to fuel extrahepatic tissues, particularly the brain. Values in (**A–H**) are expressed as the mean ± standard deviation (SD) ($n$ = 5 mice per group in **A–G** and $n$ = 7 mice per group in **H**) and were analyzed statistically by employing unpaired Student's *t*-tests (*p < 0.05, **p < 0.01, ***p < 0.001, ****p < 0.0001, n.s., no significant difference).

The online version of this article includes the following figure supplement(s) for figure 8:

**Figure supplement 1.** Accumulation of acetate derivatives in muscle is retarded under energy stress.

**Figure supplement 2.** Analyses of the behavior of diabetic mice with KD of ACOT12 or ACOT8.

for muscle force and neuromuscular coordinated movement ability. The EPMT test, correct alteration in YMZT, and object recognition index in the NOR test showed no significant difference among normal mice, diabetic mice, and Acot12/8 KD mice (*Figure 8—figure supplement 2C, D, G*), indicating that psychiatric abilities, memory, and cognitive behaviors are not markedly influenced by Acot12/8 KD in the early stage of diabetes mellitus. This is possibly because the brain exhibits the greatest flexibility among all extrahepatic tissues under energy stress in utilizing all available energy sources, including glucose, ketone bodies and acetate.

## Discussion

It is well known that glucose and ketone bodies are the main fuels for the brain in different physiological conditions. In normal conditions, the brain uses glucose as the main energy source. By contrast, it utilizes ketone bodies as an important energy source when in an energy stress status, such as diabetes mellitus or prolonged starvation, because in these cases body glucose stores have already been already exhausted and gluconeogenesis cannot provide sufficient glucose. As a result, sustained mobilization of stored lipid and the resultant production of ketone bodies from FAO are crucial for nourishing extrahepatic tissues, in particular the brain in energy stress conditions. However, sustained FAO in hepatocytes requires the rapid recycling of free CoA, which is the crucial co-enzyme for FAO. Our study elucidates that this requirement is satisfied by ACOT12- and ACOT8-catalized conversion of acetyl-CoA to acetate and CoA. It is important to point out that the significance of ACOT12- and ACOT8-catalyzed recycling of free CoA for FAO is analogous to that of lactate dehydrogenase-catalyzed recycling of NAD$^+$ for glycolysis (*Castro et al., 2009*; *Cerdán et al., 2006*). In addition to providing CoA, this reaction also provide acetate that serves as an energy source to fuel extrahepatic tissues (*Figure 8I*). As an alternative fuel in emergency status, acetate may be preferred by the brain because it is directly converted to acetyl-CoA by acetyl-CoA synthetase (ACSS), and is undoubtedly more convenient than acetoacetate and 3-HB, which need two and three enzyme-catalyzed steps, respectively, to be converted to acetyl-CoA. In this regard, we consider acetate as an emerging novel 'ketone body', and recommend that its blood level should be detected, together with those of classic ketone bodies, to indicate the status of FAO in liver and of lipid mobilization in adipose tissue upon energy stress.

In summary, this study clarified where and how acetate is produced in energy stress conditions and identifies the profound biological significance of acetate production catalyzed by ACOT12 and ACOT8. More importantly, we suggest that acetate is an emerging novel 'ketone body' and may be used as a parameter to evaluate the progression of energy stress in the future.

## Materials and methods
### Collection of clinical samples

Human clinical serum samples were collected with the ethical approval of the clinical research ethics committee of The First Affiliated Hospital of Xiamen University (Xiamen, Fujian, China). Information describing the patients (diagnosed with type II diabetes) and healthy volunteers is provided in *Figure 1—source data 1*. The serum samples were stored in a −80°C refrigerator and were mainly obtained from The First Affiliated Hospital of Xiamen University (China) after obtaining informed consent.

## Animal studies

All animal studies were approved by the Animal Ethics Committee of Xiamen University (China) (acceptance no: XMULAC20190166). BALB/c and C57BL/6 mice (6–7 weeks, random sex, in groups) were obtained from Xiamen University Laboratory Animal Center (China). C57BLKS/J-LepR$^{db}$/LepR$^{db}$ (db/db) mice were purchased from GemPharmatech Co, Ltd (China). All animals were kept in SPF conditions with 12 hr light–dark cycle, free chow and water accessed to standard rodent diet in accordance with institutional guidelines. For STZ-induced diabetic models (*Gonzalez et al., 2003*; *Like and Rossini, 1976*), mice were randomized and fasted for 12 hr but water was allowed before intraperitoneal injection of STZ (a single high dose of 150 mg/kg). Note that BALB/c diabetic mice induced by STZ needed to be fed with a 60 kcal% fat diet (high fat diet, HFD) for acetate detection until the end of the experiment. For the animal starvation experiment, mice were fasted but water was allowed. For the antibiotic treatment experiment, mice were treated with a mixture of antibiotics including 1 mg/ml ampicillin, 5 mg/ml streptomycin, and 1 mg/ml colistin in sterile drinking water for 3 weeks before STZ injection or a starvation experiment, and treatment was continued to the end of experiment (*Vétizou et al., 2015*). Cre-Loxp-mediated liver-specific Acot12 or Acot8 knockout mice (*Acot12*$^{-/-}$ or *Acot8*$^{-/-}$), C57BL/6JGpt-*Acot12*$^{em1Cflox}$/Gpt and C57BL/6JGpt-*Acot8*$^{em1Cflox}$/Gpt mice were purchased from GemPharmatech Co, Ltd (China). *Acot12* $^{Flox/Flox}$ or *Acot8* $^{Flox/Flox}$ mice crossed with *Alb-Cre* mice (C57BL/6), confirmed the efficient deletion of Acot12 or Acot8 specifically in the liver (6–7 weeks, random sex, in groups). For adenovirus-mediated liver-specific RNAi mouse models, injection of 200 μl adenovirus (titer: 10$^{12}$) via the tail vein was performed in C57BL/6 mice (6–7 weeks, random sex, in groups) and knockdown efficiency in the liver was determined by Western Blot 2 months after the injection. Adenoviruses were propagated in QBI-293A cells and purified by cesium chloride density gradient ultracentrifugation. For all animal models, blood was collected from the tail vein, followed by detection of glucose with a Roche glucometer and other serum components with NMR and MS. Mouse tissue samples were collected at the end of experiment for Western Blot.

## Animal behavior analysis

Behavior analysis was performed by using age-matched C57BL/6 mice (4 months, random sex, in groups). Diabetes mellitus was induced by STZ in normal or adenovirus-mediated liver-specific RNAi mice. Mice were moved to the experimental room 1 hr before starting the experiment. All objects or pieces of apparatus were thoroughly cleaned with 75% alcohol between trials to remove odors. All mice exhibited excellent health throughout the study period.

For the forelimb grip force test, the forelimb grip forces were measured using a Grip Strength Meter (Ugo Basile, Italy) and the peak force was defined as the average of three successive measurements. The rotarod test was performed using a progressive acceleration setting from 5 to 40 rpm for 2 min using a five-lane apparatus (Ugo Basile, Italy). Before the rotarod test, all mice were trained at 5 rpm for 2 min daily for 2 days. For acetate-rescuing experiments, the forelimb grip force test and rotarod test were conducted after 5 min of intraperitoneal injection of acetate.

The Elevated Plus Maze Test (EPMT) was performed with a maze of 40 cm in length, 10 cm in width, and 50 cm in height (Panlad, Spain), which consists of four elevated arms radiating from a central platform, forming a plus shape. Two of the opposed arms were enclosed by a wall of 20 cm in height. Each mouse was placed in the same area, and then left to explore the maze for 5 min. The amount of time spent in the open and closed arms was measured with a video-imaging system (Dazzle DVC100 Video). Data analyses were performed using active-monitoring software (smart3.0). The Y-Maze Test (YMZT) was carried out using a Y-shaped maze with three light-colored, opaque arms (30 cm in length, 6 cm in width, and 5 cm in height; Panlad, Spain) orientated at 120 angles from each other. Each mouse was placed in the same area, and then left to explore the maze for 5 min. The number of entries into the arms and alterations were recorded with a video-imaging system (Dazzle DVC100 Video). Data analyses were performed using active-monitoring software (smart3.0).

The one-trial Novel Object Recognition (NOR) test was carried out using an open-field apparatus (40 × 40 × 40 cm; Panlad, Spain) as the test box. The protocol consisted of two test sessions separated by a delay of over 20 min, during which the mice were returned to their home cage. In each test session, every mouse was placed in the same area, and then left to explore the open field for 5 min. In the first session, the mice were trained in the arena where two cubes (5 × 5 × 5 cm, familiar object) were placed as objects A and B. In the second session, the mice were trained in the arena where one

object A and one cylinder (5 cm diameter, 5 cm height, novel object), designated as object C, were placed. The time and distance of novel and familiar objects exploration were recorded with a video-imaging system (Dazzle DVC100 Video) during each session. Data analyses were performed using active-monitoring software (smart3.0). The ratio of object C exploration time to total time represents the object recognition index.

## Plasmids constructs

Full-length cDNAs encoding human ACOTs (gene ID: 25082 for ACOT1, gene ID: 15824 for ACOT2, gene ID: 9637 for ACOT4, gene ID: 24012 for ACOT8, gene ID: 17595 for ACOT9, gene ID: 10617 for ACOT11, and gene ID: 134526 for ACOT12) were obtained from the Core Facility of Biomedical Sciences, Xiamen University. Point mutations of ACOT12 (ACOT12 R312E and R313E) (*Lu et al., 2019*) and ACOT8 (ACOT8 H78A) (*Ishizuka et al., 2004*) were constructed by PCR-mediated mutagenesis using PrimerSTAR DNA polymerase (Takara). cDNAs for proteins expression were constructed in pLV cs2.0 vectors. shRNAs were constructed in lentivirus-based pLL3.7 vectors. shRNAs against mouse *Acot12* (#5, #6, and #7) and mouse *Acot8* (#1 and #2) were also constructed in pAdEasy-1 (Stratagene) for adenovirus packaging based on The AdEasy Technology (*He et al., 1998*).

## Cell culture, transfections, and cell treatments

HeLa, HEK-293T, HT1080, Huh7, LO$_2$, H3255, A549, QBI-293A, and HEB cell lines were taken from our laboratory cell bank and authenticated by Short Tandem Repeat (STR) profiling analysis. HCT116, 786-O, HepG2, Hepa1-6, and AML12 cell lines were obtained from and authenticated by the Cell Bank of the Chinese Academy of Sciences (Shanghai). All cells tested negative for mycoplasma infection using a PCR-based Mycoplasma Detection Kit (Sigma, MP0035-1KT).

Mouse embryonic fibroblast cells were isolated from the embryos of mice at 13.5 days post-coitum and further immortalized by infection with the SV-40 larger T antigen expressing retroviruses. All of the cell lines were cultured in DMEM (Gibco) with 10% fetal bovine serum (FBS, Gemini) at 37°C in an incubator containing 5% CO$_2$. HEK-293T was used for transient transfection and lentivirus package with polyethylenimine (10 μM, Polyscience) as a transfection reagent. The virus-containing medium was collected after 24 hr of transfection, filtered with a 0.45-μm Steriflip filter (Millipore) and stored at −80°C for infection. The infected cells were passaged until stable cell lines were constructed. For all kinds of treatment, cells were seeded in 35 mm dishes and cultured for 24 hr before treatment. To measure glucose-derived acetate, cells were rinsed with phosphate-buffered saline (PBS) and then incubated in glucose-free DMEM (1 ml, Gibco) supplemented with 10% FBS and 10 mM U-$^{13}$C-glucose for 20 hr before harvest. To measure fatty-acid-derived acetate, cells were rinsed with PBS and then incubated in HBSS supplemented with 10% FBS and 500 μM bovine serum albumin (BSA, fatty acids free, Yeasen Biotech)-conjugated FFA (myristate, palmitate, stearate, or U-$^{13}$C-palmitate) for 20 hr before harvest. To determine amino-acid-derived acetate, cells were rinsed with PBS and then incubated in HBSS supplemented with 10% FBS and 2× or 4× amino acids for 20 hr before harvest. 2× amino acids contains double concentrations of amino acids in DMEM and 4× contains quadruple concentrations of amino acids in DMEM. HBSS media (1 l, pH 7.4) contains CaCl$_2$ (140 mg), MgCl$_2$·6H$_2$O (100 mg), MgSO$_4$·7H$_2$O (100 mg), KCl (400 mg), KH$_2$PO$_4$ (60 mg), NaHCO$_3$ (350 mg), NaCl (8 g), and Na$_2$HPO$_4$ (48 mg). To determine gluconeogenesis, cells were rinsed with PBS and then incubated in HBSS media (glucose-free) supplemented with 100 nM glucagon (Acmec, G78830) for 4 hr before harvest as previously described (*Liu et al., 2017*).

## Adenovirus packaging and infection

Sterile linearized recombinant AdEasy plasmids were transfected into QBI-293A cell lines with Turbofect transfection reagent for adenovirus packaging as described previously (*He et al., 1998*). Fresh QBI-293A cells were further infected by the primary adenoviruses for amplification and purification of recombinant adenovirus. The purified adenovirus was used for infection of MPHs *in vitro* and liver cells *in vivo*.

## Isolation of mouse primary hepatocytes

MPHs were obtained from C57BL/6 mice by perfusing the liver through the portal vein with calcium-free buffer A (1 mM EGTA (ethylene glycol-bis(β-aminoethyl ether)-N,N,N′,N′-tetraacetic acid) and

Kreb-Ringer buffer), followed by perfusion with buffer B (collagenase-IV from Sigma, Kreb-Ringer buffer and 5 mM $CaCl_2$). Hepatic parenchymal cells were maintained in DMEM containing 10% FBS and precipitated by centrifugation (50 g for 3 min). Next, the isolated cells were plated in dishes that were pre-treated with collagen-I (CORNING) and cultured in DMEM with 10% FBS at 37°C in a humid incubator containing 5% $CO_2$. Kreb-Ringer buffer (1 l, pH 7.4) contains NaCl (7 g), $NaHCO_3$ (3 g), HEPES (4-(2-hydroxyethyl)-1-piperazineethanesulfonic acid, 5 mM, pH 7.45), Solution C (10 ml), and glucose (1 g). Solution C contains KCl (480 mM), $MgSO_4$ (120 mM), and $KH_2PO_4$ (120 mM). All of the buffers and media described above contained penicillin (100 IU, Sangon Biotech) and streptomycin (100 mg/ml, Sangon Biotech) (*Huang et al., 2011*).

## Immunoprecipitation and western blot

Cells or tissues were harvested in a lysis buffer (20 mM Tris–HCl (pH 7.4), 150 mM NaCl, 1 mM EDTA (ethylenediaminetetraacetic acid), 2.5 mM sodium pyrophosphate, 1 mM β-glycerolphosphate, 1 mM sodium orthovanadate, 1 mM EGTA, 1% Triton, 1 μg/ml leupeptin, 1 mM phenylmethylsulfonyl fluoride), sonicated and centrifuged at 20,000 × *g* for 15 min at 4°C. For immunoprecipitation, the supernatant was incubated with the corresponding antibody for 12 hr at 4°C and then incubated with A/G plus-agarose beads (Santa Cruz Biotechnology, Inc) for 2 hr at 4°C. For western blot, immuno-precipitates or total cell lysate supernatant was added to sodium dodecyl sulfate (SDS) loading buffer, boiled for 10 min and separated by SDS–polyacrylamide gel electrophoresis, followed by transfer to PVDF (polyvinylidene fluoride) membranes (Roche). The PVDF membranes were incubated with specific antibodies for 3 hr and the proteins were visualized using an enhanced chemiluminescence system. The intensity of the blots was analyzed using ImageJ.

## Subcellular fraction purification

For subcellular fraction purification, the Peroxisome Isolation kit (Sigma) was used to isolate peroxisomes from primary hepatocytes by referring to the protocol provided by Sigma-Aldrich. The isolated subcellular fractions were lysed with lysis buffer and analyzed by western blot.

## Immunofluorescence

$LO_2$ cells grown on coverslips at 30%–40% of confluence were washed with PBS and fixed in 4% para-formaldehyde for 10 min. The fixed cells were treated with 0.2% Triton X-100 in PBS for 10 min at room temperature to permeabilize the membrane and then incubated with 5% BSA in TBST (20 mM Tris, 150 mM NaCl, and 0.1% Tween 20) for 1 hr to block non-specific binding sites. Next, the cells were incubated with primary antibodies diluted in TBST containing 5% BSA for 1 hr at room temperature and washed three times with 0.02% Triton X-100 in PBS, followed by incubation with fluorescent secondary antibodies for 1 h. After washing three times with 0.02% Triton X-100 in PBS, all coverslips were counterstained with DAPI (4',6-diamidino-2-phenylindole) and mounted on microscope slides with 90% glycerol. Images were captured using a Leica TCS SP8 confocal microscope at pixels of 1024 × 1024.

## Biochemical analyses

To prepare mouse serum, mouse blood was collected into 1.5 ml Eppendorf tubes and allowed to clot for 30 min at 4°C. Then, samples were centrifuged for 30 min (1300 × *g*) at 4°C and the serum layer was carefully moved into a new 1.5 ml Eppendorf tube. Plasma levels of TG, CHOL, HDL-C, and LDL-C were measured in the Clinical Laboratory of Zhongshan Hospital, which is affiliated to Xiamen University. Plasma insulin levels were measured using the MOUSE INS-1055 ELISA KIT (Meikebio) with a standard curve, following the manufacturer's protocol. Plasma total FFAs levels were measured using the FFA content assay kit (Beijing Boxbio Science & Technology) with a standard curve, following the manufacturer's protocol, and each FFA was measured by gas chromatography — mass spectrometry (GC–MS). Fasted mice were fasted for 12 hr; non-fasted mice were fed normally.

## Gas chromatography–mass spectrometry

To identify the acetate produced by cells, metabolites in the culture medium were subjected to acidification and extraction, followed by analysis using GC–MS as previously described but with some optimization (*Fellows et al., 2018*). First, equal amounts of propionic acid and butyric acid were added

to cell-cultured media (100 μl) in Eppendorf tubes as an internal reference. Subsequently, 40 mg of sodium chloride, 20 mg of citric acid and 40 μl of 1 M hydrochloric acid were added to acidize the metabolites. After acidification, acetate, propionic acid, and butyric acid were liable to be extracted by 200 μl of n-butanol. Next, the tubes were vortexed for 3 min and centrifuged at 20,000 × $g$ for 20 min. The supernatant was transferred to a high-performance liquid chromatography (HPLC) vial and 1 μl mixture was used for GC-MS.

For measurement of FFA levels, mouse serum was extracted as mentioned above and subjected to GC-MS. In brief, 30 μl of cold mouse serum was transferred to new 1.5 ml Eppendorf tubes and 500 μl of cold 50% methanol (containing 2.5 μg/ml tridecanoic acid as internal reference) was added to the samples, followed by the addition of 500 μl of cold chloroform. Next, samples were vortexed at 4°C for 10 min and centrifuged (12,000 × $g$) at 4°C for 20 min to separate the phase. The chloroform phase containing the total fatty acid content was separated and lyophilized by nitrogen. Dried fatty acid samples were esterified with 100 μl 1% sulfuric acid in methanol for 60 min at 80°C and extracted by the addition of 100 μl n-hexane. The supernatant was transferred to a HPLC vial and 1 μl mixture was used for GC-MS.

Analysis was performed using an Agilent 7890B gas chromatography system. This system was coupled to an Agilent 5977B mass spectrometric detector and a fused-silica capillary DB-FFAP, with dimensions of 30 m × 0.25 mm internal diameter (i.d.), coated with a 0.25-μm thick layer. The initial oven temperature was 50°C, which was ramped to 110°C at a rate of 15°C min$^{-1}$, to 180°C at a rate of 5°C min$^{-1}$, to 240°C at a rate of 15°C min$^{-1}$, and finally held at 240°C for 10 min. Helium was used as a carrier gas at a constant flow rate of 1 ml min$^{-1}$ through the column. The temperatures and the mass spectral data were collected in a full scan mode ($m/z$ 30–300).

## Liquid chromatography–mass spectrometry

The metabolites of TCA cycle were determined by LC–MS as described (*Hui et al., 2020*). To prepare the samples for the measurement of metabolites in tissues, the equivalent tissues of brain or muscle were quenched by a pre-cold methanol solution (methanol:ddH$_2$O = 4:1) and homogenized, followed by centrifugation (12,000 × $g$, 20 min). The supernatants were collected in new Eppendorf tubes and dried at 4°C before the pellets were resuspended in acetonitrile solution (acetonitrile:ddH$_2$O=1:1) and transferred to a HPLC vial. To prepare the samples for the measurement of the intracellular metabolites of *in vitro* cultured cells, the medium was discarded and cells cultured in a 35 mm dish were gently washed twice with cold PBS, followed by the addition of pre-cold methanol solution (methanol:ddH$_2$O = 4:1, containing 160 ng/ml U-$^{13}$C-glutamine as internal reference) to each well. Samples were then handled as described above for the measurement of tissue metabolites. For the analysis of metabolites the samples prepared as described above, liquid chromatography was performed using a SCIEX ExionLC AD system, with all chromatographic separations using a Millipore ZIC-pHILIC column (5 μm, 2.1 × 100 mm internal dimensions, PN: 1.50462.0001). The column was maintained at 40°C and the injection volume of all samples was 2 μl. The mobile phase, which consisted of 15 mM ammonium acetate and 3 ml/l ammonium hydroxide (>28%) in LC–MS grade water (mobile phase A) and LC–MS grade 90% (vol/vol) acetonitrile in HPLC water (mobile phase B), ran at a flow rate of 0.2 ml/min. The ingredients were separated with the following gradient program: 95% B for 2 min, then changed to 45% B within 13 min (linear gradient) and maintained for 3 min, then changed to 95% B directly and maintained for 4 min. The flow rate was 0.2 ml/min. The QTRAP mass spectrometer used an Turbo V ion source, which was run in negative mode with a spray voltage of −4500 V, with Gas1 40 psi, Gas2 50 psi, and Curtain gas 35 psi. Metabolites were measured using the multiple reactions monitoring mode. The relative amounts of metabolites were analyzed using MultiQuant Software Software (AB SCIEX).

## NMR measurements

To prepare the culture medium samples for NMR analysis, medium harvested after the treatment of cells were centrifuged (12,000 × $g$ at 4°C for 10 min) and the supernatants (400 μl) were transferred into 5 mm NMR tubes for NMR measurement. The clinical serum samples (200 μl) were thawed on ice, mixed with 200 μl NMR buffer (50 mM sodium phosphate buffer, pH 7.4 in D$_2$O) and centrifuged (12,000 × $g$) at 4°C for 10 min. The supernatants (400 μl) were transferred into 5 mm NMR tubes for NMR measurement. For preparation of mouse blood samples, 25 μl blood was mixed with 75 μl

saline immediately, and centrifuged (3000 × $g$) at 4°C for 10 min. The supernatants (100 μl) were mixed with 300 μl NMR buffer and transferred into 5 mm NMR tubes for NMR measurement. An internal tube containing 200 μl $D_2O$ (used for field-frequency lock) with 1 mM sodium 3-(trimethylsilyl) propionate-2,2,3,3-d4 (TSP) was used to provide the chemical shift reference (δ 0.00) and to quantify the metabolites.

NMR measurements were performed on a Bruker Avance III 850 MHz spectrometer (Bruker BioSpin, Germany) equipped with a TCI cryoprobe at 25°C provided by the College of Chemistry and Chemical Engineering (Xiamen University) and a Bruker Avance III 600 MHz spectrometer (Bruker BioSpin, Germany) provided by the Core Facility of Biomedical Sciences (Xiamen University). One-dimensional (1D) CPMG spectra were acquired using the pulse sequence [RD-90°-($\tau$-180°-$\tau$)$_n$-ACQ] with water suppression for culture medium and serum samples. For the purpose of metabolite resonance assignments, two-dimensional (2D) $^1H$-$^{13}C$ heteronuclear single quantum coherence spectra were recorded on selected NMR samples. Identified metabolites were confirmed by a combination of 2D NMR data and the Human Metabolome Data Base (HMDB).

## FAO measurement

FAO measurement was carried out as previously described (*Li et al., 2018*). Briefly, cells were cultured in 35 mm dishes and rinsed twice with PBS to remove the residue medium. 1 ml reaction buffer (119 mM NaCl, 10 mM HEPES (pH 7.4), 5 mM KCl, 2.6 mM $MgSO_4$, 25 mM $NaHCO_3$, 2.6 mM $KH_2PO_4$, 2 mM $CaCl_2$, 1 mM BSA-congregated oleic acid and 0.8 μCi/ml [9,10-$^3H(N)$]-oleic acid) was added to each dish and cells were incubated at 37°C for 12 hr, followed by centrifugation (1000 × $g$, 5 min) to obtain a supernatant. Then, 192 μl of 1.3 M perchloric acid was added to 480 μl of supernatant. The mixture was centrifuged (20,000 × $g$, 5 min) and 240 μl of the resulting supernatant was mixed with 2.4 ml of scintillation liquid and 3H radioactivity, followed by measurement with a liquid scintillation counter (Tri-Carb 2008TR, Perkins Elmer, USA; provided by the Center of Major Equipment and Technology (COMET), State Key Laboratory of Marine Environmental Science, Xiamen University).

## Database analysis

The RNA-Seq data were downloaded from the GSE72086 dataset (*Goldstein et al., 2017*) of the public NCBI Gene Expression Omnibus (https://www.ncbi.nlm.nih.gov/geo/), analyzed using the limma package (*Ritchie et al., 2015*), and visualized using the ggplot2 and ggrepel packages in R (version 3.6.3). GSE72086 contains six samples: three fed treatments and three fasted 24 hr treatments. Here, the probe with the greatest p value was chosen to determine the differential gene expression for multiple probes corresponding to the same gene. Adjusted p value <0.05 and |log fold change (log FC) | ≥ 1 were chosen as the threshold values.

The tissue-specific mRNA expression of the target gene was analyzed using the GTEx database (https://www.gtexportal.org/home/) for the human data and GSE24207 of the GEO database for the mice data, as reported previously (*Fagerberg et al., 2014*; *Thorrez et al., 2011*).

## Statistical analysis

The two-tailed Student's *t*-test was performed, using GraphPad Prism 8 and Excel, to analyze the difference between two groups. One-way analysis of variance was performed using GraphPad Prism 9 and R (version 3.6.3) to compare values among more than two groups. Difference was considered significant if the p value was lower than 0.05 (*p < 0.05; **p < 0.01; ***p < 0.001; ****p < 0.0001).

## Acknowledgements

We thank members of the Qinxi Li's laboratory for productive discussions and comments on this manuscript. We thank the Center of Major Equipment and Technology (COMET), State Key Laboratory of Marine Environmental Science (Xiamen University) for technical support.

# Additional information

## Funding

| Funder | Grant reference number | Author |
| --- | --- | --- |
| National Natural Science Foundation of China | U21A20373 | Qinxi Li |
| National Natural Science Foundation of China | 91857102 | Qinxi Li |
| National Natural Science Foundation of China | 31701252 | Bin Jiang |
| National Natural Science Foundation of China | 32100601 | Huanhuan Ma |
| Open Research Fund of State Key Laboratory of Cellular Stress Biology | SKLCSB2020KF002 | Qinxi Li |
| Open Research Fund of State Key Laboratory of Cellular Stress Biology | SKLCSB2019KF005 | Qinxi Li |

The funders had no role in study design, data collection, and interpretation, or the decision to submit the work for publication.

## Author contributions

Jinyang Wang, Conceptualization, Data curation, Software, Formal analysis, Supervision, Validation, Investigation, Visualization, Methodology, Writing - original draft, Project administration, Writing - review and editing; Yaxin Wen, Data curation, Validation, Investigation, Visualization; Wentao Zhao, Conceptualization, Investigation, Writing - original draft; Yan Zhang, Furong Lin, Conceptualization, Investigation; Cong Ouyang, Huihui Wang, Lizheng Yao, Yue Zhuo, Investigation; Huanhuan Ma, Funding acquisition; Huiying Huang, Conceptualization, Methodology; Xiulin Shi, Investigation, Methodology; Liubin Feng, Methodology; Donghai Lin, Resources, Methodology; Bin Jiang, Conceptualization, Funding acquisition, Methodology, Project administration; Qinxi Li, Conceptualization, Resources, Funding acquisition, Writing - original draft, Writing - review and editing

## Author ORCIDs

Jinyang Wang ⬚ http://orcid.org/0000-0002-2517-7473
Wentao Zhao ⬚ http://orcid.org/0000-0003-4787-5245
Huanhuan Ma ⬚ http://orcid.org/0000-0001-8029-6814
Qinxi Li ⬚ http://orcid.org/0000-0002-7275-4595

## Ethics

This study was approved by the Clinical Research Ethics Committee of the First Affiliated Hospital of Xiamen University (Xiamen, Fujian, China). Before clinical serum samples acquisition, written informed consent was obtained from each patient according to the policies of the committee.

All animal studies were approved by the Animal Ethics Committee of Xiamen University (China) (acceptance no: XMULAC20190166). All surgery was performed under sodium pentobarbital anesthesia, and every effort was made to minimize suffering.

Reviewer #1 (Public Review): https://doi.org/10.7554/eLife.87419.3.sa1
Reviewer #2 (Public Review): https://doi.org/10.7554/eLife.87419.3.sa2
Reviewer #3 (Public Review): https://doi.org/10.7554/eLife.87419.3.sa3
Author Response https://doi.org/10.7554/eLife.87419.3.sa4

# Additional files

## Supplementary files
• MDAR checklist

## Data availability

All data generated or analyzed during this study are included in the manuscript and supporting file. Source data files have been provided for Figures 1, 3, 4, 5, and 7. The numerical data utilized for the generation of the figures has been duly submitted to the Mendeley Data repository (https://doi.org/10.17632/xvs8292ytz.1).

The following dataset was generated:

| Author(s) | Year | Dataset title | Dataset URL | Database and Identifier |
|---|---|---|---|---|
| Wang J, Li Q | 2023 | Hepatic conversion of acetyl-CoA to acetate plays crucial roles in energy stresses | https://doi.org/10.17632/xvs8292ytz.1 | Mendeley Data, 10.17632/xvs8292ytz.1 |

The following previously published datasets were used:

| Author(s) | Year | Dataset title | Dataset URL | Database and Identifier |
|---|---|---|---|---|
| Thorrez L, Laudadio I, Van Deun K, Quintens R, Hendrickx N, Granvik M, Lemaire K, Schraenen A, Van Lommel L, Lehnert S | 2010 | mRNA analysis in different mouse tissues | https://www.ncbi.nlm.nih.gov/geo/query/acc.cgi?acc=GSE24207 | NCBI Gene Expression Omnibus, GSE24207 |
| Goldstein I, Baek S, Presman DM, Paakinaho V, Swinstead EE, Hager GL | 2016 | Characterization of chromatin and gene expression changes during fasting in mouse liver [RNA-Seq] | https://www.ncbi.nlm.nih.gov/geo/query/acc.cgi?acc=GSE72086 | NCBI Gene Expression Omnibus, GSE72086 |

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

# Appendix 1

**Appendix 1—key resources table**

| Reagent type (species) or resource | Designation | Source or reference | Identifiers | Additional information |
|---|---|---|---|---|
| Gene (*Homo-sapiens*) | ACOT1 | Core Facility of Biomedical Sciences, Xiamen University | Gene ID: 25082 | |
| Gene (*Homo-sapiens*) | ACOT2 | Core Facility of Biomedical Sciences, Xiamen University | Gene ID: 15824 | |
| Gene (*Homo-sapiens*) | ACOT4 | Core Facility of Biomedical Sciences, Xiamen University | Gene ID: 9637 | |
| Gene (*Homo-sapiens*) | ACOT8 | Core Facility of Biomedical Sciences, Xiamen University | Gene ID: 24012 | |
| Gene (*Homo-sapiens*) | ACOT9 | Core Facility of Biomedical Sciences, Xiamen University | Gene ID: 17595 | |
| Gene (*Homo-sapiens*) | ACOT11 | Core Facility of Biomedical Sciences, Xiamen University | Gene ID: 10617 | |
| Gene (*Homo-sapiens*) | ACOT12 | Core Facility of Biomedical Sciences, Xiamen University | Gene ID: 134526 | |
| Cell line (*Homo-sapiens*) | HeLa | Our laboratory cells bank | | Cell line maintained in our laboratory cells bank |
| Cell line (*Homo-sapiens*) | HEK-293T | Our laboratory cells bank | | Cell line maintained in our laboratory cells bank |
| Cell line (*Homo-sapiens*) | HT1080 | Our laboratory cells bank | | Cell line maintained in our laboratory cells bank |
| Cell line (*Homo-sapiens*) | Huh7 | Our laboratory cells bank | | Cell line maintained in our laboratory cells bank |
| Cell line (*Homo-sapiens*) | LO$_2$ | Our laboratory cells bank | | Cell line maintained in our laboratory cells bank |
| Cell line (*Homo-sapiens*) | H3255 | Our laboratory cells bank | | Cell line maintained in our laboratory cells bank |
| Cell line (*Homo-sapiens*) | A549 | Our laboratory cells bank | | Cell line maintained in our laboratory cells bank |
| Cell line (*Homo-sapiens*) | QBI-293A | Our laboratory cells bank | | Cell line maintained in our laboratory cells bank |
| Cell line (*Homo-sapiens*) | HEB | Our laboratory cells bank | | Cell line maintained in our laboratory cells bank |
| Cell line (*Homo-sapiens*) | HCT116 | Cell Bank of the Chinese Academy of Sciences (Shanghai) | | Cell line maintained in Cell Bank of the Chinese Academy of Sciences (Shanghai) |
| Cell line (*Homo-sapiens*) | 786-O | Cell Bank of the Chinese Academy of Sciences (Shanghai) | | Cell line maintained in Cell Bank of the Chinese Academy of Sciences (Shanghai) |
| Cell line (*Homo-sapiens*) | HepG2 | Cell Bank of the Chinese Academy of Sciences (Shanghai) | | Cell line maintained in Cell Bank of the Chinese Academy of Sciences (Shanghai) |

*Appendix 1 Continued on next page*

*Appendix 1 Continued*

| Reagent type (species) or resource | Designation | Source or reference | Identifiers | Additional information |
|---|---|---|---|---|
| Cell line (*mouse*) | Hepa1-6 | Cell Bank of the Chinese Academy of Sciences (Shanghai) | | Cell line maintained in Cell Bank of the Chinese Academy of Sciences (Shanghai) |
| Cell line (*mouse*) | AML12 | Cell Bank of the Chinese Academy of Sciences (Shanghai) | | Cell line maintained in Cell Bank of the Chinese Academy of Sciences (Shanghai) |
| Transfected construct (*human*) | ACLY shRNA-#1 | This paper | | Lentiviral construct to transfect and express the shRNA; Targeting sequence: GCAGCAGACCTATGACTATGC |
| Transfected construct (*human*) | ACLY shRNA-#2 | This paper | | Lentiviral construct to transfect and express the shRNA; Targeting sequence: GCATCGCAAACTTCACCAACG |
| Transfected construct (*human*) | ACLY shRNA-#3 | This paper | | Lentiviral construct to transfect and express the shRNA; Targeting sequence: GCACGAAGTCACAATCTTTGT |
| Transfected construct (*human*) | ACLY shRNA-#4 | This paper | | Lentiviral construct to transfect and express the shRNA; Targeting sequence: GCAAGGCATGCTGGACTTTGA |
| Transfected construct (*human*) | CPT1A shRNA-#3 | This paper | | Lentiviral construct to transfect and express the shRNA; Targeting sequence: TACAGTCGGTGAGGCCTCTTATGAA |
| Transfected construct (*human*) | CPT1A shRNA-#4 | This paper | | Lentiviral construct to transfect and express the shRNA; Targeting sequence: GGACCAAGATTACAGTGGTATTTGA |
| Transfected construct (*human*) | ABCD1 shRNA-#1 | This paper | | Lentiviral construct to transfect and express the shRNA; Targeting sequence: GCAGATCAACCTCATCCTTCT |
| Transfected construct (*human*) | ACOT12 shRNA-#1 | This paper | | Lentiviral construct to transfect and express the shRNA; Targeting sequence: GCTAGAGTTGGACAAGTTATA |
| Transfected construct (*human*) | ACOT12 shRNA-#2 | This paper | | Lentiviral construct to transfect and express the shRNA; Targeting sequence: CAAATACCAGTGATTTGGATTAGCA |
| Transfected construct (*mouse*) | Acot12 shRNA-#5 | This paper | | Lentiviral construct to transfect and express the shRNA; Targeting sequence: GCATGGAGATCAGTATCAAGG |
| Transfected construct (*mouse*) | Acot12 shRNA-#6 | This paper | | Lentiviral construct to transfect and express the shRNA; Targeting sequence: GCAGGTTCAGCGATTCCATTT |
| Transfected construct (*mouse*) | Acot12 shRNA-#7 | This paper | | Lentiviral construct to transfect and express the shRNA; Targeting sequence: GCGAGGACGATCAGATATATT |
| Transfected construct (*human*) | ACOT8 shRNA-#1 | This paper | | Lentiviral construct to transfect and express the shRNA; Targeting sequence: GAGGATCTCTTCAGAGGAAGG |
| Transfected construct (*human*) | ACOT8 shRNA-#2 | This paper | | Lentiviral construct to transfect and express the shRNA; Targeting sequence: GCAGCCAAGTCTGTGAGTGAA |
| Transfected construct (*mouse*) | Acot8 shRNA-#1 | This paper | | Lentiviral construct to transfect and express the shRNA; Targeting sequence: GGGACCCTAACCTTCACAAGA |

*Appendix 1 Continued on next page*

*Appendix 1 Continued*

| Reagent type (species) or resource | Designation | Source or reference | Identifiers | Additional information |
|---|---|---|---|---|
| Transfected construct (*mouse*) | Acot8 shRNA-#2 | This paper | | Lentiviral construct to transfect and express the shRNA; Targeting sequence: GCTGTGTGGCTGCTTA TATCT |
| Antibody | anti-Flag (mouse monoclonal) | Sigma | Cat#F1804 RRID:AB_262044 | IF (1:200) WB (1:2000) |
| Antibody | anti-ACOT12 (rabbit polyclonal) | Abbkine | Cat#ABP53776 | WB (1:500) |
| Antibody | anti-ACOT8 (rabbit polyclonal) | Abbkine | Cat#ABP50586 | WB (1:500) |
| Antibody | anti-HMGCS2 (rabbit polyclonal) | ABclonal | Cat#A14244 RRID:AB_2761104 | WB (1:1000) |
| Antibody | anti-ABCD1 (rabbit polyclonal) | Abbkine | Cat#ABP54187 | WB (1:1000) |
| Antibody | anti-β-actin (mouse monoclonal) | Proteintech | Cat#60008-1-Ig RRID:AB_2289225 | WB (1:2000) |
| Antibody | anti-ACLY (rabbit polyclonal) | Proteintech | Cat#15421-1-AP RRID:AB_2223741 | WB (1:500) |
| Antibody | anti-CPT1A (rabbit polyclonal) | Proteintech | Cat#15184-1-AP RRID:AB_2084676 | WB (1:500) |
| Antibody | anti-catalase (mouse monoclonal) | Proteintech | Cat#66765-1-Ig RRID:AB_2882111 | IF (1:100) |
| Antibody | anti-TOMM40 (mouse monoclonal) | Proteintech | Cat#66658-1-Ig RRID:AB_2882015 | WB (1:2000) |
| Antibody | anti-LAMP2 (mouse monoclonal) | Proteintech | Cat#66301-1-Ig RRID:AB_2881684 | WB (1:2000) |
| Antibody | anti-GAPDH (rabbit monoclonal) | Proteintech | Cat#60004-1-Ig RRID:AB_2107436 | IF (1:100) WB (1:2000) |
| Antibody | Acetylated-lysine antibody (rabbit polyclonal) | Cell Signaling Technology | Cat#9441S RRID:AB_331805 | WB (1:1000) |
| Antibody | HRP-conjugated goat anti-mouse IgG antibody | Thermo Fisher | Cat#A16072SAM PLE | WB (1:5000) |
| Antibody | HRP-conjugated goat anti-rabbit IgG antibody | Thermo Fisher | Cat#A16104SAM PLE | WB (1:5000) |
| Commercial assay or kit | Peroxisome Isolation kit | Sigma | PEROX1-1KT | |
| Commercial assay or kit | PCR-based Mycoplasma Detection Kit | Sigma | MP0035-1KT | |
| Chemical compound, drug | Streptozotocin | Sangon Biotech | Cat#A610130-0100 | |
| Chemical compound, drug | Ampicillin | Sangon Biotech | Cat#A610028-0025 | |
| Chemical compound, drug | Streptomycin | Sangon Biotech | Cat#A610494-0250 | |
| Chemical compound, drug | Tetradecanoic acid | Sangon Biotech | Cat#A600931-0250 | |

*Appendix 1 Continued on next page*

*Appendix 1 Continued*

| Reagent type (species) or resource | Designation | Source or reference | Identifiers | Additional information |
|---|---|---|---|---|
| Chemical compound, drug | Sodium stearate | Sangon Biotech | Cat#A600888-0100 | |
| Chemical compound, drug | Colistin | Yuanye Bio-Technology | Cat#1264-72-8 | |
| Chemical compound, drug | Deuterated water (D$_2$O) | Qingdao Tenglong Weibo Technology | Cat#DFSA180309 G100 | |
| Chemical compound, drug | Sodium 3-(trimethylsilyl) propionate-2,2,3,3-d4 (TSP) | Qingdao Tenglong Weibo Technology | Cat#DLM-48–5 | |
| Chemical compound, drug | Etomoxir | MedChemExpress (MCE) | Cat#828934-41-4 | |
| Chemical compound, drug | Sodium palmitate | Sigma | Cat#P9767-10G | |
| Chemical compound, drug | Sodium acetate | Sigma | Cat#791741-100G | |
| Chemical compound, drug | Sodium 3-hydroxybutyrate | Sigma | Cat#54965-10G-F | |
| Chemical compound, drug | U-$^{13}$C-palmitate | Cambridge Isotope Laboratories | Cat#CLM-6059–1 | |
| Chemical compound, drug | U-$^{13}$C-glucose | Cambridge Isotope Laboratories | Cat#CLM-1396-1 | |
| Chemical compound, drug | U-$^{13}$C-glutamine | Cambridge Isotope Laboratories | Cat#CLM-1822-H-0.1 | |
| Chemical compound, drug | U-$^{13}$C-acetate | Cambridge Isotope Laboratories | Cat#CLM-440-1 | |
| Chemical compound, drug | 2-$^{13}$C-acetate | Cambridge Isotope Laboratories | Cat#CLM-381-5 | |
| Chemical compound, drug | DAPI stain | Sigma | D9542 | |
| Software, algorithm | R | R-studio | | R (version 3.6.3) |

