## [Editor Report · eLife assessment]

This is **important** work that examines hepatic acetate production via ACOT12/18 in starvation and diabetes. The investigators use **solid** loss of function strategies in cells, including mouse primary hepatocytes, and in vivo mouse experiments to show that ACOTs are necessary for normal acetate production in the context of fasting and type 1 diabetes. Given that acetate is commonly thought to primarily represent a fermentation product, this study is of interest as it describes hepatic pathways converting fatty acids to acetate.

---

## [Referee Report · Reviewer #1 (Public Review)]

The authors investigate the roles of ACOT12/8 in the production of acetate by the liver. They observe that acetate concentration parallels ketone concentrations during fasting and T1DM. They show that acetate is produced from fatty acids in hepatocytes, but though described as a novel "ketone body", this acetate is not a product of ketogenesis or acetoacetate. They also provide serum acetate data from human subjects who were classified as either "healthy" or "diabetic,". These subjects are noted as T2DM patients, but there is no other characterization or description, making it difficult to ascertain the context in which they were studied or their relevance to the mouse studies. Although the function of ACOT12/8 is reported in the literature, they are not widely studied, and there also remains surprising uncertainties regarding the mechanism of acetate production by the liver. In this regard, the manuscript provides some important insight. The authors use ShACOT12/8 and ACOT12/8 knockout mice to demonstrate that these acetyl-CoA hydrolases are largely necessary for acetate production. Using a 3H-palmitate assay, the authors then find that loss of these ACOTs inhibit fatty acid oxidation and propose that the mechanism involves scavenging CoA, analogous to the canonical role of ketogenesis. The idea is plausible but not proven. A related finding is that loss of these ACOTs inhibit ketogenesis, which the authors attribute to the loss of function of HMGC2S, partially through acetylation. These mechanisms suffer some limitations based on the cytosolic and mitochondrial compartmentation of the two processes, but the observations appear sound. Interestingly, the loss of the ACOTs have a more profound effect on lowering ketones than acetate, which may have parallel effects but they are not investigated. Finally, the authors try to demonstrate that hepatic ACOT-mediated acetate production is necessary for normal motor function in STZ treated mice, ostensibly as compensation for impaired glucose utilization by the CNS. Injections of 13C acetate and 13C enrichment in downstream metabolites of brain are used to support the importance of acetate metabolism, but the experiment was not performed in loss of function models. In addition, the resulting 13C enrichment data is reported generically as "relative intensity" without further elaboration on how this data was generated and should not be taken at face value by the reader. Conceptually, one may also be skeptical of the rather dramatic loss of motor function in the context of a relatively minor circulating nutrient. Nevertheless, this finding may be important if more supporting evidence with proper controls for ketone concentrations can be provided. Overall, there are important data in the manuscript, but the reader may find it difficult to navigate the 20+ figure panels. The most important findings are that ACOT12/8 are critical for hepatic acetate production in mice, which will be helpful for the field, but the ramifications require more rigorous investigation.

---

## [Referee Report · Reviewer #2 (Public Review)]

Catabolic conditions lead to increased formation of ketone bodies in the liver, which under these conditions play an important role in supplying energy to metabolically active organs. In this manuscript, the authors explore the concept of whether and to what extent hepatic formation of acetate might contribute to energy supply under metabolic stress conditions. The authors show that patients with diabetes have increased acetate levels, which is explained as a consequence of the increased fatty acid flux from adipose tissue to the liver. This is confirmed in a preclinical model for type 1 diabetes, where acetate concentrations are in a similar range to ketone bodies. Acetate concentrations also increase under physiological conditions of fasting. Using stable isotopes, the authors show that palmitate is used as the primary source for acetate production in primary hepatocytes. Using cell culture studies and adenoviral-mediated knockdown in mice, it can be shown that the conversion of acetyl-CoA to acetate is catalyzed in peroxisomes by acyl-CoA thioesterase8 (ACOT8) and after transport of citrate from mitochondria and subsequent conversion to acetyl-CoA in the cytosol by ACOT12. Remarkably, ACOT8/12 not only regulates the formation of acetate but plays a crucial role in the maintenance of cellular CoA concentration. Accordingly, depletion of ACOT8/12 activity leads to a reduction of other CoA derivatives such as HMG-CoA, which resulted in the inhibition of ketone body synthesis. In diabetic mice, ACOT 8 or ACOT12 knockdown appears to lead to some limitations in strength and behavior.

In summary, the authors clearly demonstrate that hepatic release-mediated by ACOT8 and ACOT12-determines the plasma concentration of acetate. This is a very remarkable observation since most studies assume that short-chain fatty acids in plasma are primarily generated by fermentation of dietary fiber by intestinal bacteria. The authors demonstrate in very well performed studies the metabolic changes that result from impaired thiolysis. On the other hand, the ACOT12 phenotype has been demonstrated in a recently published study (PMID: 34285335). In this study, ACOT12 deficiency caused NAFLD, thus it would be worth determining whether deficiency of ACOT12 and/or ACOT8 promotes de novo lipogenesis under the conditions of the present study. As a further limitation, it should be noted that the relevance of acetate production for the energy supply of peripheral organs including the central nervous system could not be clearly demonstrated. For instance, impaired ketone body production due to impaired CoA availability could affect the metabolic activity of various organs. Moreover, the human cohort is not very well described, e.g. it is unclear whether the patients have type 1 or type 2 diabetes.

---

## [Referee Report · Reviewer #3 (Public Review)]

Wang et al. investigated the role of acetate production, a byproduct of fatty acid oxidation, in the context of metabolic stressors, including diabetes mellitus and prolonged fasting. Mechanistically, they show the importance of the liver enzymes ACOT8 (peroxisome) and ACOT12 (cytoplasm) in converting FFA-derived acetyl-CoA into acetate and CoA. The regeneration of CoA allows for subsequent fatty acid oxidation. Inhibiting the generation of acetate has negative motor consequences in streptozocin-treated mice, which are mitigated with acetate injection.

This paper's strengths include using multiple mouse models, metabolic stressors (db/db-/-, streptozocin, and prolonged starvation), numerous cell lines, precise knockout and rescue experiments, and complimentary use of mass spectrometry and nuclear magnetic resonance analytical platforms. The presented data support the conclusions of this paper and highlight the role of acetate in energy stress conditions.

In clinical medicine, common ketones that are measured are acetoacetate, beta-hydroxybutyrate, and acetone which can help determine the severity of illness. However, the data presented here suggest the potential importance of measuring acetate as another biomarker when patients present with ketoacidosis in uncontrolled diabetes or starvation. This requires further investigation.

---

## [Author Response]

The following is the authors’ response to the original reviews

**Reviewer #1 (Recommendations for the authors):**
Major Concerns:1. There are numerous grammatical issues throughout the manuscript, and too much awkward jargon is used, such as "status of energy stresses", "ES-acetate". The characterization of acetate as an "energy stress" gives a negative connotation, which is unnecessary and confusing. Ketones are produced under the same circumstances but are a vital adaptive response, except for ketoacidosis. The terminology used throughout the manuscript is also vague, and some methodology is not adequately described in the Methods section. For example, the meaning of "preprandial" and "postprandial" is unclear, and there is no explanation of the related methodology.

Thank you for your comments. We have replaced "status of energy stresses" with "energy stresses", in our revised manuscript. We agree with you that acetate and Ketone Bodies are produced under the same circumstances and their production is a result of a vital adaptive response. It is well known that the production of large amount of acetate and Ketone Bodies is an important physiological adaption of body in response to energy stresses such as prolonged starvation and untreated diabetes mellitus. In this context, we use “energy stress-acetate”, a term coined by ourselves to emphasize the condition of acetate production and its role under such condition. Based on your concerns, we have addressed the issues and provided a thorough description of the modifications made in the Methods section.

1. The authors claim that acetate is a ketone body, which is incorrect. As the authors show, it is not produced by the ketogenic pathway or from the breakdown of ketones. Acetate is a carboxylic acid and specifically a short-chain fatty acid.

We agree with you that our description of acetate as a ketone body is seemingly incorrect. Indeed, acetate is a short-chain fatty acid in terms of molecular structure. The classic Ketone Bodies include acetone, acetoacetate and beta-hydroxybutyrate, among which acetone and acetoacetate contain carbonyl group and can be considered as ketone, however beta-hydroxybutyrate which contains only hydroxyl and carboxyl groups is actually not a ketone but a short-chain fatty acid.Noteworthily, here our description of acetate as an emerging novel “ketone body” is not aimed to consider it as a real ketone in structure, but to emphasize the high similarity of acetate and the classic Ketone Bodies in the organ (liver) and substrate (fatty acids-derived acetyl-CoA) of their production, the roles they played (as important sources of fuel and energy for many extrahepatic peripheral organs), the feature of their catabolism (converted back to acetyl-CoA and degraded in TCA cycle), as well as the physiological conditions of their production (energy stresses such as prolonged starvation and untreated diabetes mellitus). To prevent any potential misunderstanding, we annotate the usage of "ketone body" with double quotation marks in our revised manuscript.

1. The human subjects are not sufficiently characterized, and it is unclear whether they are T1DM or T2DM subjects. No information is provided on morphometrics, how and when serum was collected, exclusion criteria, medicines, etc. Proper characterization of human subjects is necessary before publishing such data.

Thank you very much for your comments. We have added the description of subjects you mentioned in the Methods section.

1. While Figure 4 is an essential set of experiments that demonstrate that ACOT12 is necessary for the induction of acetate during starvation in mice, the authors do not explain the source of basal levels of acetate that persist in mice lacking ACOT12. It is unclear whether this source is from other tissue or microbiota. Since loss of ACOT by ShRNA treatment resulted in ~25% reduction in acetate, it is very difficult to conceive how this produces the profound neurological and strength deficits presented in Supplemental Figure 8 (see last point below).Additionally, it is not clear how the control mice for the knockout studies were generated. Please clarify.

In normal condition, the serum acetate level in mice is around 200 μM. Hepatic ACOT12 and ACOT8 enzymes seems to provide a serum acetate concentration of 60-90 μM, individually (Figure 4). The intestinal microbiota contributes a serum acetate concentration of 60-80 μM (Figure 2 and Figure supplement 1).

During energy stress, the protein levels of ACOT12 and ACOT8 in the mouse liver were significantly upregulated (Figure 3 and Figure supplement 1), resulting in an significant increase of serum acetate level to approximate 400 μM. The acetate produced by ACOT12 (~200 μM) and ACOT8 (~200 μM) constitutes the main portion of serum acetate concentration under such condition (Figure 2), while the contribution of intestinal microbiota to serum acetate level is minimized (Figure 2 and Figure supplement 1). Elimination of either ACOT12 or ACOT8 reduces serum acetate level by up to 50% (Figure 4). However, such estimation is only a rough approximation and does not consider the possibility of compensatory upregulation of ACOT12 and ACOT8 in kidney when ACOT12 or ACOT8 is knocked out in liver.

Acetate assumes the role as an important energy source in the case of reduced glucose utilization associated with diabetes. In this case, knockdown of ACOT12 or ACOT8 (shACOT12 or shACOT8) can remarkably reduce acetate production and consequently influence the Motor Function of mice to a certain extent.

1. The results presented in Figure 5 are confusing, and the authors' interpretation needs elaboration. The FAO assay detects water-soluble 3H-metabolites and 3H2O, and etimoxir or CPT1 knockout completely inhibits FAO. Therefore, it is unclear how peroxisomes can produce acetate without generating water-soluble intermediates that are detectable in the assay. Further explanation and rationale for the authors' interpretation are necessary.

Mitochondria serve as the primary organelle for the catabolism of oleic acid. However, in certain instances, fatty acid oxidation (FAO) can occur in the peroxisome, resulting in the production of medium-chain fatty acids and acetyl-CoA. Nevertheless, these medium-chain fatty acids cannot undergo further oxidation within the peroxisome. Instead, they must be transported out of the peroxisome and then into the mitochondria through CPT1 (carnitine palmitoyltransferase 1) for further oxidation.

To assess FAO, we utilized a detection method based on 3H labeling in H2O in cells treated with [9,10-3H(N)]-oleic acid. The introduction of [9,10-3H(N)]-oleic acid leads to the production of 3H-labeled medium-chain fatty acids and acetyl-CoA within the peroxisome. The further oxidation of 3H-labeled medium-chain fatty acids in the mitochondria was inhibited by impeding the activity of CPT1, leading to the eventual decrease of 3H-labeled H2O. However, acetyl-CoA can still be converted to acetate by ACOT8. As a result, knockdown or etomoxir inhibition of CPT1, decreased more than one-half of U-13C-palmitate-derived U-13C-acetate production, in spite of mitochondria β-oxidation being nearly completely abolished.

1. Figure 6F, which shows various fatty acyl-CoAs in MPHs, is not helpful on its own. It would be useful to compare this data to loss of function MPH data and to measure these acyl-CoAs in knockout liver. Additionally, since it is normal for liver acetyl-CoA concentration to change by several-fold in fasted and fed liver, this data from snap frozen liver tissue of ACOT12/8 KO mice would help prove the authors' point.

We are grateful for your valuable advice. As you mentioned there are indeed several outstanding questions that require further clarification. To address these questions, we are currently in the process of developing an experimental mouse model in which ACOT12 and ACOT8 are conditionally knocked out. By virtue of this approach, we aim to acquire more substantial evidence to substantiate the aforementioned conclusions.

1. Figure 7 suggests that loss of ACOT inhibits ketogenesis by decreasing HMGCS2 expression and increasing its acetylation. However, it is difficult to imagine that this the main mechanism considering the extraordinary ability of liver to handle high rates of acetyl-CoA conversion to ketones during fasting which, as the authors know, is the canonical mechanism by which mitochondrial CoA is preserved during elevated FAO. The manuscript (Figure 6 and 7) argues that it is the conversion of acetyl-CoA to acetate which is more important. A critical limitation of this argument is that ACOT12 is in cytosol (Figure 5), so while it spares CoA for fatty acid activation, it does not spare CoA for beta oxidation in mitochondria. That latter function is carried out by the ketogenic pathway. A second limitation is that the mechanism relies on citrate transport and ACLY activity, which is not generally thought to be very active in the ketogenic states of fasting and T1DM studied here. In essence, the mechanism relies on circular logic, whereby mitochondrial acetyl-CoA accumulates in the setting of impaired FAO, which then impairs ketogenesis and depletes CoA which then impairs FAO without lowering acetyl-CoA. I don't have a solution, but I think it is important to acknowledge the flaws in this proposed mechanism.

As the Reviewer suggested, ACLY indeed plays a crucial role in fatty acid synthesis. Acetyl-CoA is transported out of the mitochondria in the form of citrate, which is subsequently broken down into acetyl-CoA by ACLY. Under conditions of sufficient nutrition, acetyl-CoA carboxylase 1 further activates acetyl-CoA to participate in fatty acid synthesis.

In the context of an energy crisis resulting from low glucose utilization, we propose that ACLY might serve another pivotal role in addressing this energy deficit. In conditions such as untreated diabetes or prolonged starvation, glucose utilization is significantly reduced, leading to a reliance of body on fatty acid oxidation in liver to generate Ketone Bodies and acetate to fuels extrahepatic peripheral tissues and thus cope with the energy crisis. However, excessive fatty acid oxidation disrupts the balance between oxidized and reduced CoA, necessitating the production of both acetate and Ketone bodies to restore this equilibrium. Conventionally, fatty acid synthesis is inhibited during this period as AMPK is activated to suppress acetyl-CoA carboxylase 1 activity via phosphorylation in low-energy states. Based on our preliminary experimental results, the activity of ACLY and citrate transporter still appear to work well. It is possible that citrate-ACLY-ACOT12-acetate pathway is important for downregulating the level of mitochondria acetyl-CoA in energy crisis. According to previous studies, cytosolic reduced CoA has the capability to be transported into the mitochondria, thereby replenishing the acetyl-CoA pool within the mitochondria (PMID: 32234503). It is important to note that this remains a hypothesis requiring further testing.

1. Figure 8 presents some deceptively complex MS data following a 13C-acetate injection. The data is presented in an unorthodox manner, as 13C-metabolite intensities, making it nearly impossible to properly interpret. Enrichment of TCA cycle intermediates are not always easy to interpret, but at minimum, this data needs to be presented as MIDs or fractional enrichments. If the data is not modeled, then it might be useful to at least perform a rudimentary precursor-product analysis (i.e. normalized to plasma acetate enrichment).Supplemental Figure 8 also introduces evidence for neurological and strength deficits in shACOT12/8 knockdown mice. It is an interesting observation, but there is no direct link to the metabolic studies in the main figure, which does not present data in the loss of function mice. Nor is this part of the story investigated in liver specific knockout mice. Figure 8 is the least developed part of the manuscript and could be removed without losing the impact of the story.

We deeply appreciate your valuable suggestions. As mentioned previously, we are currently engaged in the development of an experimental mouse model where ACOT12 and ACOT8 are selectively knocked out. Subsequent experiments will be conducted to validate this model, and the resulting data will be presented in the form of MIDs or fractional enrichments, as per your suggestion.

The evaluation of anxiety-related behavior is commonly done using the Elevated Plus Maze Test (EPMT), while working memory and cognitive functions are assessed through the Y-maze Test (YMZT) and Novel Object Recognition (NOR) Test. Measures such as forelimb strength and running time in the rotarod test, total distance in YMZT, total entries in YMZT, and total distance in the NOR test are indicators of muscle force and movement ability. Our data demonstrate that acetate plays a significant role in enhancing muscle force and facilitating coordinated neuromuscular movement. Interestingly, we found that ACOT12/8 knockdown in the early stages of diabetes mellitus does not have a pronounced impact on psychiatric, memory, and cognitive behaviors (Figure 8 and figure supplement 2). However, it is important to note that our study primarily focuses on elucidating the utilization of acetate during energy crises, such as untreated diabetes and chronic hunger. Our findings suggest that acetate is primarily utilized to enhance motor capacity rather than cognitive or neural activity.

**Reviewer #2 (Recommendations for the authors):**
The statement that acetate is an emerging ketone body is not correct. It is not a ketone, it is a carboxylic acid or a short-chain fatty acid. In my opinion, to avoid confusion this should be clarified.

We agree with you that our description of this is not clear enough. Acetate is a short-chain fatty acid in terms of molecular structure indeed.

The classic Ketone Bodies include acetone, acetoacetate and beta-hydroxybutyrate, among which acetone and acetoacetate contain carbonyl group and can be considered as ketone, however beta-hydroxybutyrate which contains only hydroxyl and carboxyl groups is actually not a ketone but a short-chain fatty acid.

Noteworthily, here our description of acetate as an emerging novel “ketone body” is not aimed to consider it as a real ketone in structure, but to emphasize the high similarity of acetate and the classic Ketone Bodies in the organ (liver) and substrate (fatty acids-derived acetyl-CoA) of their production, the roles they played (as important sources of fuel and energy for many extrahepatic peripheral organs), the feature of their catabolism (converted back to acetyl-CoA and degraded in TCA cycle), as well as the physiological conditions of their production (energy stresses such as prolonged starvation and untreated diabetes mellitus). To prevent any potential misunderstanding, we annotate the usage of "ketone body" with double quotation marks in our revised manuscript.

The reason for increased fatty acid delivery to the liver is explained by insulin resistance rather than by reduced carbohydrate availability.Patient characteristics should be provided.

Thank you for your suggestions. We have revised our manuscript accordingly.

**Reviewer #3 (Recommendations for the authors):**
• Please include the rationale for having data from both C57BL/6 and BALC/c. In metabolic research, C57BL/6 is more commonly studied. The data between these two strains are similar, and one could be easily removed to limit redundancy.

Thank you for bringing this issue to our attention in the manuscript. In metabolic research, C57BL/6 mice are more commonly utilized as a model organism than BALC/c mice indeed. In this study we try to elucidate a characteristic may be shared among different mammalian species, namely the ability to produce a substantial amount of acetate during energy crises. However, given the constraints of our experimental setup, we opted to employ C57BL/6 mice as the main animal model to investigate the underlying mechanism. BALC/c mice were used to confirm the underlying mechanisms governing acetic acid production.

• In the experiments where ACOT8 and ACOT12 are selectively knocked out or knocked down, please include the levels of other ketone bodies, such as 3-HB and AcAC, from these experiments. While acetate production is diminished, there might or might not be a compensatory increase in the production of these metabolites. This would include experiments related to Figures 3, 4, and 5.

Thank you for your valuable comments. As you mentioned, in diabetic mice where ACOT12 and ACOT8 are knocked down in liver, there is a significant down-regulation of 3-HB and AcAc (Figure 7B, C). Based on this observation, we hypothesize that ACOT12 and ACOT8 might also play a regulatory role in the formation and metabolism of ketone bodies during an energy crisis. However, the precise regulatory mechanism underlying this phenomenon requires further investigation.

• From Figure 1 (source data 1), two patients with diabetes have concurrent cancer. Cancer cells have altered metabolism compared to native cells. Thus, it is possible that circulating acetate cells may be altered in these cancer patients, regardless of the presence of diabetes. This should be acknowledged. Otherwise, these two subjects should be taken out.

Thank you for your suggestions. We have taken out these two subjects in our revised manuscript.

• Can the authors expand on their thoughts on why some results from the behavioral tests are statistically significant while others are not? For example, many motor tasks such as forelimb strength, running time, total distance, and total entries significantly differ with ACOT8 and ACOT12 knockdown. However, more anxiety-based measures such as time in open arms, correct alteration, and object recognition are not statistically different.

Thank you for your comments. The evaluation of anxiety-related behavior is commonly done using the Elevated Plus Maze Test (EPMT), while working memory and cognitive functions are assessed through the Y-maze Test (YMZT) and Novel Object Recognition (NOR) Test. Measures such as forelimb strength and running time in the rotarod test, total distance in YMZT, total entries in YMZT and total distance in the NOR test are indicators of muscle force and movement ability. Our data demonstrate that acetate plays a significant role in enhancing muscle force and facilitating coordinated neuromuscular movement. Interestingly, we found that ACOT12/8 knockdown in the early stages of diabetes mellitus does not have a pronounced impact on psychiatric, memory, and cognitive behaviors (Figure 8 and figure supplement 2). However, it is important to note that our study primarily focuses on elucidating the utilization of acetate during energy crises, such as untreated diabetes and chronic hunger. Our findings suggest that acetate is primarily utilized to enhance motor capacity rather than cognitive or neural activity.